# A peptide-based viral inactivator inhibits Zika virus infection in pregnant mice and fetuses

Yufeng Yu[1,*], Yong-Qiang Deng[2,*], Peng Zou[1,*], Qian Wang[1], Yanyan Dai[1], Fei Yu[1], Lanying Du[3], Na-Na Zhang[2], Min Tian[4], Jia-Nan Hao[2], Yu Meng[1], Yuan Li[1], Xiaohui Zhou[1], Jasper Fuk-Woo Chan[5], Kwok-Yung Yuen[5], Cheng-Feng Qin[2], Shibo Jiang[1,3] & Lu Lu[1]

Zika virus (ZIKV), a re-emerging flavivirus associated with neurological disorders, has spread rapidly to more than 70 countries and territories. However, no specific vaccines or antiviral drugs are currently available to prevent or treat ZIKV infection. Here we report that a synthetic peptide derived from the stem region of ZIKV envelope protein, designated Z2, potently inhibits infection of ZIKV and other flaviviruses in vitro. We show that Z2 interacts with ZIKV surface protein and disrupts the integrity of the viral membrane. Z2 can penetrate the placental barrier to enter fetal tissues and is safe for use in pregnant mice. Intraperitoneal administration of Z2 inhibits vertical transmission of ZIKV in pregnant C57BL/6 mice and protects type I or type I/II interferon receptor-deficient mice against lethal ZIKV challenge. Thus, Z2 has potential to be further developed as an antiviral treatment against ZIKV infection in high-risk populations, particularly pregnant women.

[1] Key Laboratory of Medical Molecular Virology of MOE/MOH, School of Basic Medical Sciences and Shanghai Public Health Clinical Center, Fudan University, Shanghai 200032, China. [2] State Key Laboratory of Pathogen and Biosecurity, Beijing Institute of Microbiology and Epidemiology, Beijing 100101, China. [3] Lindsley F. Kimball Research Institute, New York Blood Center, New York, New York 10065, USA. [4] Beijing Hospital of Traditional Chinese Medicine affiliated to Capital Medical University, Beijing 100010, China. [5] State Key Laboratory of Emerging Infectious Diseases, Department of Microbiology, The University of Hong Kong, Hong Kong 999077, China. * These authors contributed equally to this work. Correspondence and requests for materials should be addressed to C.-F.Q. (email: qincf@bmi.ac.cn) or to S.J. (email: shibojiang@fudan.edu.cn) or to L.L. (email: lul@fudan.edu.cn).

Zika virus (ZIKV), a re-emerging mosquito-borne flavivirus, is an enveloped, positive single-stranded RNA virus[1,2]. As of 20 October 2016, 73 countries and territories had reported continuing vector-borne transmission of ZIKV (http://www.who.int/emergencies/zika-virus/en). In addition to Africa and Asia-Pacific regions, the Americas, particularly South America, are areas at risk for ZIKV transmission[3]. It is concerning that ZIKV infection can cause severe congenital brain developmental abnormalities, including microcephaly[4–6], and that it is a trigger of Guillain–Barré syndrome[7,8]. Currently, no specific anti-ZIKV therapeutic or vaccine is available, thus calling for the development of effective and safe antiviral drugs and vaccines for worldwide treatment and prevention of ZIKV infection. Some small-molecule compounds, such as 7-Deaza-2′-C-Methyladenosine[9], 2′-C-methylated nucleosides[10] and ( − )-epigallocatechin gallate[11], as well as some US Food and Drug Administration-approved drugs used in clinics for other purposes, such as PHA-690509, niclosamide, bortezomib, mycophenolic acid, mefloquine, daptomycin and bromocriptine[12–14], were found to inhibit ZIKV infection. However, the safety of small-molecule drugs for pregnant women requires more extensive evaluation. 2A10G6, a murine antibody, with broad flavivirus-neutralizing activity by targeting the ZIKV envelope (E) protein could neutralize ZIKV infection *in vitro* and *in vivo*[15]. However, its relatively low efficacy and requirement for humanization are main obstacles to its further development. ZIKV-117, a human monoclonal antibody (mAb), can broadly neutralize infection of ZIKV strains[16]. However, the high cost may limit its application in developing countries, such as Brazil.

In recent years, development of peptide drugs has attracted growing attention because of their better safety and lower development cost compared to small-molecule- and antibody-based antiviral drugs. We previously identified the first highly potent anti-HIV C-peptide, SJ-2176 (refs 17,18) and the patent was licensed to Trimeris Pharmaceutical Inc. for development of the first HIV fusion inhibitor, enfuvirtide, which was approved by the US Food and Drug Administration in 2003 for the treatment of HIV infection[19,20]. Furthermore, the use of enfuvirtide, either alone or combined with other antiretroviral drugs, could successfully prevent mother-to-child HIV transmission during pregnancy[21–23], indicating that antimicrobial peptides could be developed as safe and effective antiviral therapeutics and/or prophylactics. Therefore, we aimed to design and identify an effective anti-ZIKV peptide.

A flavivirion contains 180 copies of E protein and membrane (M) protein forming 90 E dimers, which completely cover the viral surface[24]. When fusion of viral envelope with host cell membrane occurs in low pH, the conformation of E protein changes from dimeric to trimeric[25]. The stem region in the E protein proximal to the virus membrane then changes conformation from α-helix to random coil, driving the viral transmembrane anchor towards the fusion loop and stabilizing the trimeric structure of E protein[26,27]. It has been reported that the peptides derived from the stem region of dengue virus (DENV) could effectively inhibit fusion of DENV with the host cell or induce hole formation in viral membrane, resulting in the release of viral RNA genome[27–29], suggesting that the stem region can also serve as an important target for the development of anti-ZIKV drugs.

In the present study, we designed and synthesized a peptide derived from the conserved stem region of ZIKV E protein, designated Z2, and found that Z2 inhibits ZIKV infection *in vitro* and *in vivo* through inactivating ZIKV virions. We demonstrated that Z2 is able to penetrate the placental barrier to block vertical transmission of ZIKV from pregnant mice to their fetuses. These

results suggest that Z2 could be further developed as a safe and effective peptide therapeutic and prophylactic for the treatment and prevention of ZIKV infection in high-risk populations, especially in pregnant women.

## Results

**Rational design of anti-ZIKV peptide Z2.** First, we aligned the amino-acid sequence of ZIKV E protein with that of the corresponding fragment in the stem region of DENV E protein, which represents the basis for the design of anti-DENV peptides[27–29]. We next aligned these sequences with those in the E proteins of other flaviviruses, including Japanese encephalitis virus, yellow fever virus (YFV) and West Nile virus. We found that the sequence in this region is highly conserved among flaviviruses with amino-acid sequence conservation of 64 to 82% (Fig. 1a and Supplementary Fig. 1), implying that this region may play important roles in flavivirus infection. Finally, we located this region in the 3.8 Å resolution cryo-electron microscope structure of ZIKV (Protein Data Bank: 5IRE)[30], as shown in pink in Fig. 1b, and it was confirmed as the membrane-proximal stem region of ZIKV E protein (residues 421–453), and this was then used as the basis for the design and synthesis of peptide Z2 and the scrambled peptide of Z2 (Z2-scr).

**Z2 inhibited ZIKV infection at early viral replication stage.** To determine the antiviral activity of Z2 against ZIKV infection in BHK21 and Vero cells, we developed a rapid and sensitive colorimetric viral infection assay using Cell Counting Kit-8 (CCK8, Dojindo, Japan)[31–33]. It was reported that ZIKV infection of these cells resulted in obvious cytopathic effects (CPE)[34]. Using this assay, we tested the inhibitory activity of Z2 at different concentrations on infection of ZIKV strain SZ01. We found that Z2 inhibited ZIKV infection in a dose-dependent manner with a 50% inhibitory concentration (IC$_{50}$) value of $1.75 \pm 0.13\,\mu$M (mean ± s.d., $n = 3$) in BHK21 cells (Fig. 2a) and $3.69 \pm 0.27\,\mu$M ($n = 3$) in Vero cells (Fig. 2b), while Z2-scr as a negative control showed no significant inhibition on ZIKV infection. We also used the plaque reduction assay and BHK21 cells to test anti-ZIKV activity. As shown in Supplementary Fig. 2, the IC$_{50}$ value is $2.61 \pm 0.46\,\mu$M ($n = 3$), suggesting that the result derived from the plaque reduction assay is consistent with that obtained from colorimetric CCK8 assay. The ZIKV-induced CPE in BHK21 and Vero cells were visibly reduced by Z2 at $5\,\mu$M, while Z2-scr at the same concentration exhibited no inhibitory activity (Fig. 2c). In line with the data from CCK8 assay, the result from immunofluorescence staining showed that Z2 at a concentration of $10\,\mu$M almost completely blocked ZIKV E protein expression in BHK21 and Vero cells (Fig. 2d). We then evaluated the antiviral activity of Z2 against other ZIKV strains. As shown in Table 1, Z2 was also effective in inhibiting infection of ZIKV strains FLR and MR766 in BHK21 cells with IC$_{50}$ values of about 4 and $14\,\mu$M, respectively, suggesting that Z2 possesses broad inhibitory activity against ZIKV strains isolated from patients or rhesus monkeys in different regions of the world.

Results of the time-of-addition experiment (Supplementary Fig. 3) showed that Z2 inhibited ZIKV infection as early as 2 h post infection, indicating that Z2 may inhibit infection of ZIKV before its entry into the target cell. Following this, we investigated whether Z2 could also inhibit infection by other mosquito-borne flaviviruses, such as DENV and YFV. We found that Z2 was also highly effective in inhibiting infection by DENV-2 and YFV 17D (Fig. 2e) with IC$_{50}$ values of about 4 and $5\,\mu$M, respectively, while it had no inhibition on infection by pseudotyped vesicular stomatitis virus (VSV) and Middle East Respiratory Syndrome Coronavirus (Fig. 2f). These results suggest that Z2 may have

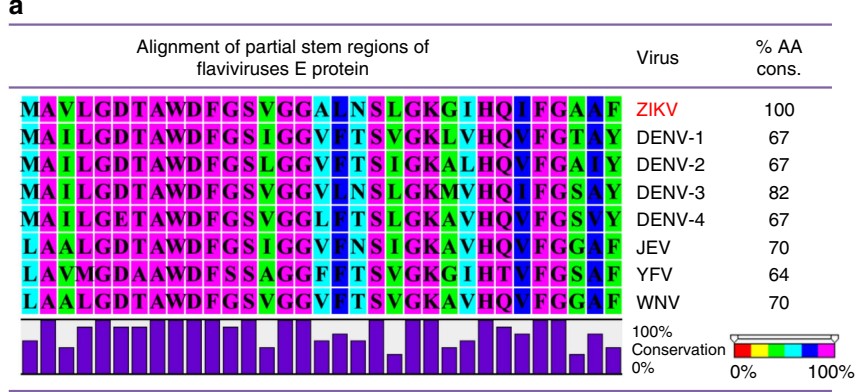

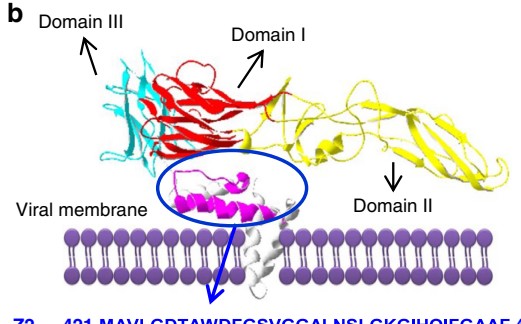

Z2 421 MAVLGDTAWDFGSVGGALNSLGKGIHQIFGAAF 453

Z2-scr LDIIAGLSAGFQGGATFVDAHGMVKASFLGGNW

**Figure 1 | Design of peptide inhibitor Z2.** (**a**) Sequence alignment of stem regions from E protein of flaviviruses. JEV, Japanese encephalitis virus; WNV, West Nile virus. The % amino-acid conservation (%AA cons.) from stem region of ZIKV is shown. (**b**) Sequence and location of Z2 in the stem region of ZIKV E protein. The structure of E protein was generated by SWISS-MODEL software based on the 3.8 Å resolution cryo-electron microscope structure of ZIKV (Protein Data Bank: 5IRE)[30]. Red, domain I of ZIKV E protein; yellow, domain II; cyan, domain III; pink, peptide Z2; purple, viral membrane. Z2-scr, scrambled peptide of Z2.

inhibitory activity against infection of a broad spectrum of flaviviruses.

**Z2 bound to E protein and inactivated ZIKV virions.** It was reported that DN59, a peptide derived from the stem region of the DENV E protein, could inhibit flavivirus infection by interacting with virus particles and inducing formation of pores in viral envelope membrane, resulting in the release of viral RNA genome[28]. Using similar approaches, we investigated whether Z2 could interact with ZIKV E protein and induce the release of RNA genome from ZIKV virions. First, we used an immunofluorescence staining assay to show that 293T cells transfected by pcDNA3.1-Env could be bound by 4G2, a mouse mAb against E protein of pan-flaviviruses[35]. Meanwhile, the 293T cells transfected by the empty vector pcDNA3.1 showed no binding, suggesting that ZIKV E protein did express on the 293T cells transfected with pcDNA3.1-Env (Fig. 3a). Using a flow cytometric analysis, we demonstrated that the 293T cells transfected with pcDNA3.1-Env could be stained by Z2-Cy5, whereas the 293T cells expressing no E protein showed only background staining (Fig. 3b). Similar results were obtained from experiment using ZIKV-infected BHK21 cells (Supplementary Fig. 4). These results indicate that Z2 is able to interact with E protein of ZIKV.

Next, we measured the potential release of viral genomic RNA from ZIKV using an RNase digestion assay[28]. As shown in Fig. 3c,d, the genomic RNA of untreated virions (0 µM Z2) or treated by Z2-scr were protected from digestion of micrococcal nuclease. However, the genomic RNA of Z2 treated virions was digested by micrococcal nuclease in a dose-responsive manner.

About 70% genomic RNA of ZIKV virions treated with 50 µM Z2 was digested by micrococcal nuclease. To further confirm the release of viral genomic RNA, ZIKV virions treated with 1% dimethylsulfoxide (DMSO), 100 µM Z2 or Z2-scr in 1% DMSO, and 1% Triton X-100, respectively, was centrifuged through a sucrose density gradient as previously described[28]. Then the amount of genomic RNA and E protein of ZIKV in each fraction was monitored by reverse-transcription quantitative PCR (RT–qPCR) and western blot (Supplementary Figs 5 and 6), respectively, and the data of western blot was further analysed by Image J software. As shown in Fig. 3e, the genomic RNA and E protein of virions treated by 1% DMSO or Z2-scr migrated to the same fractions. For example, the peaks of both genomic RNA and E protein were located at fraction No. 6, suggesting that virions treated by 1% DMSO or Z2-scr maintained intact. However, the genomic RNA and E protein of virions treated by 1% Triton X-100 migrated to different fractions, that is, peaks of genomic RNA and E protein were in fraction No. 1 and 7, respectively, indicating that virions were completely destroyed by Triton X-100. Interestingly, the peaks of genomic RNA and E protein of virions treated with Z2 were located in fraction No. 6 and 3, respectively. These results indicate that Z2 treatment may cause pore formation in the viral membrane, resulting in the release of viral genomic RNA through the pores.

Next, we determined whether the virions treated with Z2 peptide were inactivated. After incubation of ZIKV with Z2 or Z2-scr at room temperature for 2 h, we separated the treated ZIKV virions from the unbound free peptide with PEG-8000, as previously described[36] and assessed their infectivity. As shown in Fig. 3f, ZIKV virions treated with Z2 lost infectivity in a

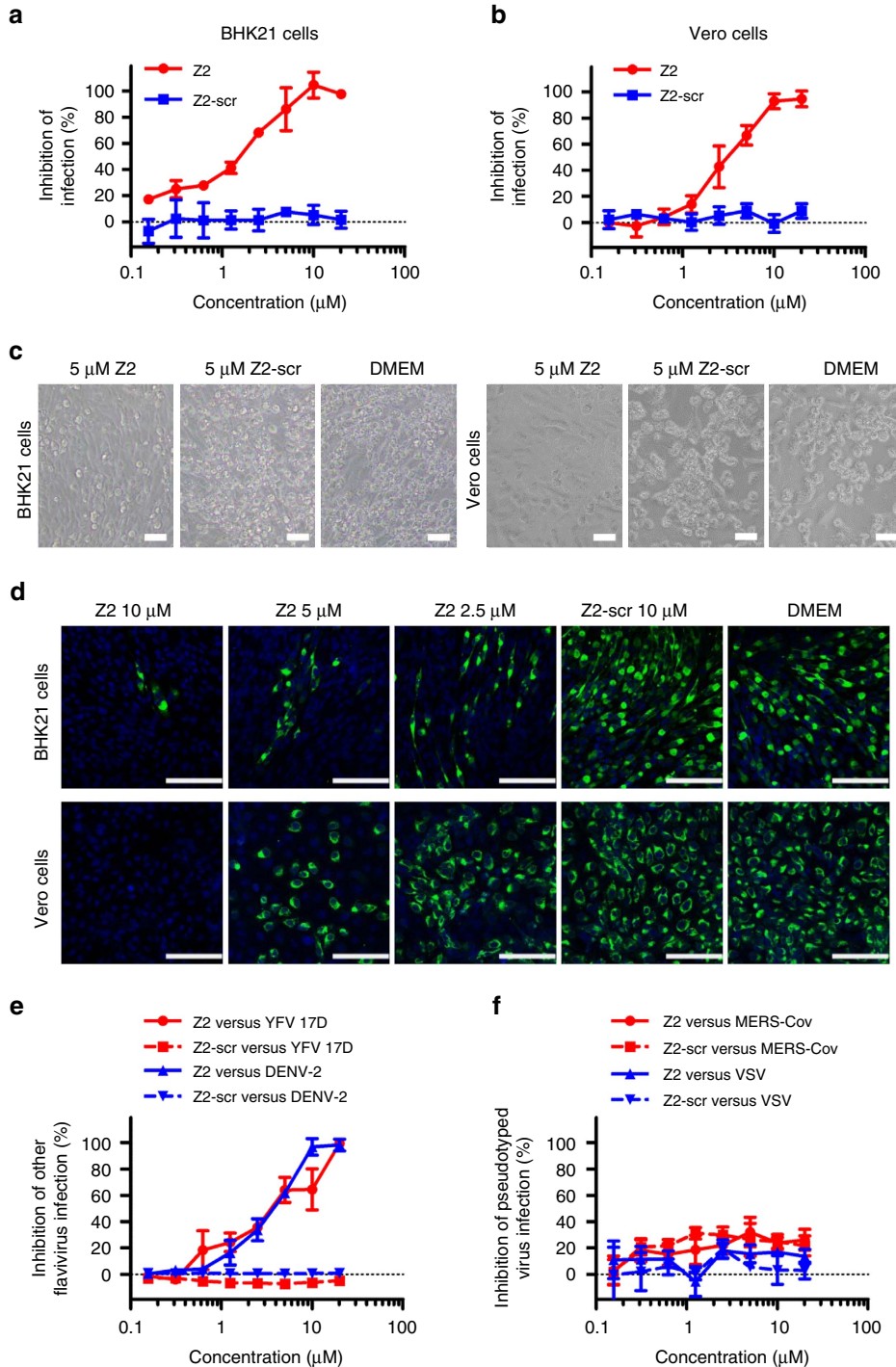

**Figure 2 | Antiviral activity of Z2 in vitro.** (**a**) Dose-dependent inhibition of ZIKV infection by Z2 in BHK21 cells. (**b**) Dose-dependent inhibition of ZIKV infection by Z2 in Vero cells. (**c**) Reduction of ZIKV-induced CPE in BHK21 and Vero cells by Z2. Scale bar, 20 μm. (**d**) Immunofluorescence assay to confirm the antiviral activity of Z2 against ZIKV. ZIKV E protein stained by the anti-E mAb 4G2 (green); nuclei stained by 4,6-diamidino-2-phenylindole (blue). Scale bar, 100 μm. (**e**) Inhibition of DENV-2 and YFV 17D in BHK21 cells by Z2. (**f**) Inhibition of pseudotyped VSV and MERS-CoV in Huh7 cells by Z2. Data are means ± s.d. of triplicate experiments.

dose-dependent manner with 50% effective concentration of $2.52 \pm 0.18 \,\mu\text{M}$ ($n = 3$), while ZIKV treated with Z2-scr at concentration as high as $50 \,\mu\text{M}$ retained 90% infectivity (Fig. 3f), suggesting that Z2-treated ZIKV virions had been inactivated.

Importantly, although Z2 could disrupt the integrity of ZIKV membranes, it showed no cytotoxic effect on cell lines (BHK21, Vero and Huh7) tested at concentration as high as $50 \,\mu\text{M}$

(Supplementary Fig. 7). Z2 also showed no cytotoxic effect on red blood cells isolated from mouse peripheral blood at concentration as high as $100 \,\mu\text{M}$ (Supplementary Fig. 8).

**Z2 could cross placental barrier of pregnant ICR mice.** ZIKV can cross the placental barrier and infect the fetus during pregnancy, causing an abnormal growth of the brain and head of

**Table 1 | Inhibition of ZIKV infection by Z2.**

| ZIKV strain | Isolated from | Site of isolation | Year of isolation | $IC_{50}$, µM mean ± s.d. ($n = 3$) |
|---|---|---|---|---|
| SZ01 | ZIKV-infected patient | China | 2016 | 1.75 ± 0.13 |
| FLR | ZIKV-infected patient | Colombia | 2015 | 4.04 ± 1.87 |
| MR766 | ZIKV-infected rhesus monkey | Uganda | 1947 | 13.91 ± 0.33 |

BHK21 cells were used to test the inhibitory activity of Z2 on ZIKV infection. $IC_{50}$ values were calculated by Graphpad Prism 5.0 software (log(inhibitor) versus response or normalized response–variable slope).

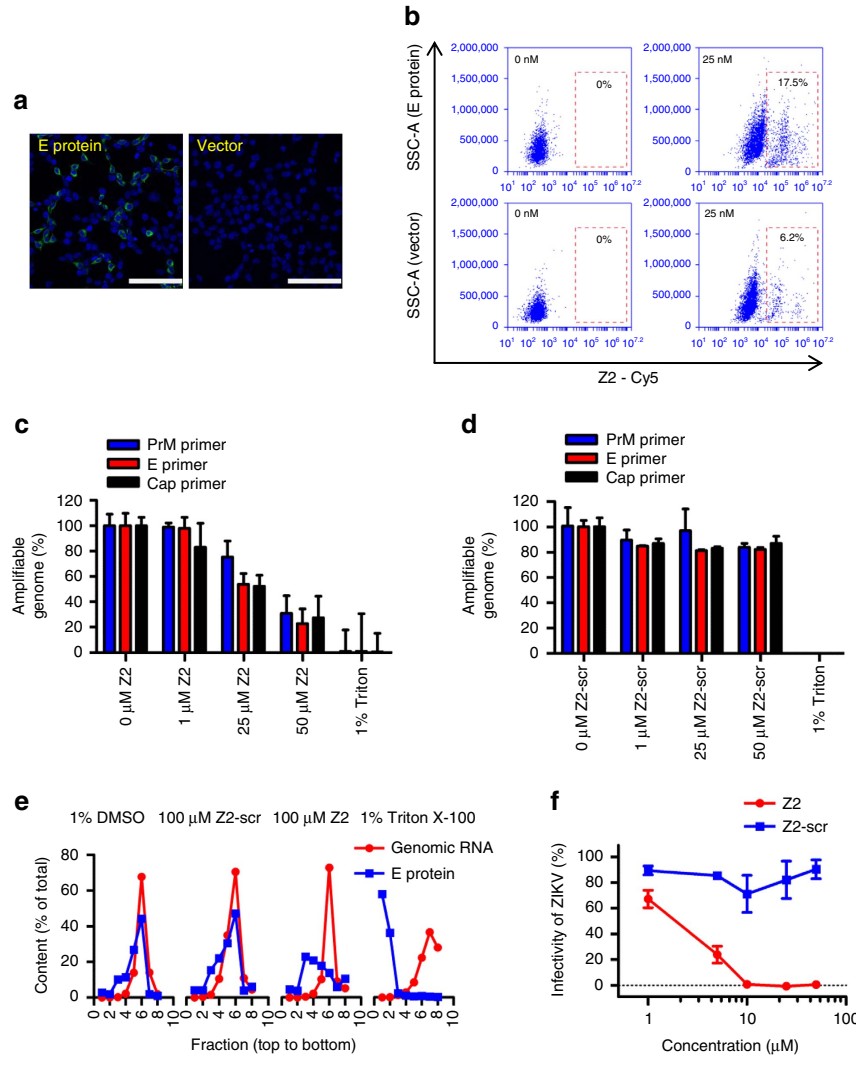

**Figure 3 | Inactivation of ZIKV by Z2.** (**a**) Immunofluorescence staining the ZIKV E protein expressed on 293T cells by the anti-E mAb 4G2 (green). Nuclei stained by 4,6-diamidino-2-phenylindole (blue). Scale bar, 100 µm. (**b**) Determination of the binding of Z2 with E protein expressed on 293T cells by flow cytometry. (**c**) Degradation of released genomic RNA of ZIKV mediated by Z2 in an RNase digestion assay. The primers were used to detect RNA sequences in viral genome coding PrM, E protein and Cap protein, respectively. (**d**) Degradation of released genomic RNA of ZIKV mediated by Z2-scr. (**e**) The separation of genomic RNA and E protein of ZIKV treated with Z2, Z2-scr, DMSO or Triton X-100 through a sucrose density gradient assay. Per cent of total E protein in each fraction was assessed by western blot and analysed by Image J software. Per cent of total RNA genome in each fraction was measured by RT–qPCR. (**f**) Inactivation of ZIKV by Z2. After incubation at room temperature for 2 h, ZIKV was separated from Z2 by PEG-8000 for the measurement of the residual infectivity. Data are means ± s.d. of triplicate experiments.

the fetus[5,37]. Therefore, an effective anti-ZIKV drug should have the ability to penetrate through the placental barrier of the infected pregnant women to protect the fetus. Here we investigate whether Z2 can enter into the pregnant mouse's organs, and penetrate through its placenta to enter into the fetus' body. As shown in Fig. 4a and Supplementary Fig. 9a,b, the fluorescent signals of Z2-Cy5 were shown at the sites where the liver and

bladder were located and the dissected organs, such as liver, kidney, spleen and heart, suggesting that Z2 is able to enter into blood circulation and organs of pregnant mice. The average radiant efficiency (p s$^{-1}$ cm$^{-2}$ sr$^{-1}$)(µW$^{-1}$ cm$^2$) in the liver ($P = 0.0275$, Student's two-tailed $t$-test) and bladder ($P = 0.0426$, Student's two-tailed $t$ test) of the Z2-treated mice was significantly higher than that in the phosphate-buffered saline (PBS)-treated

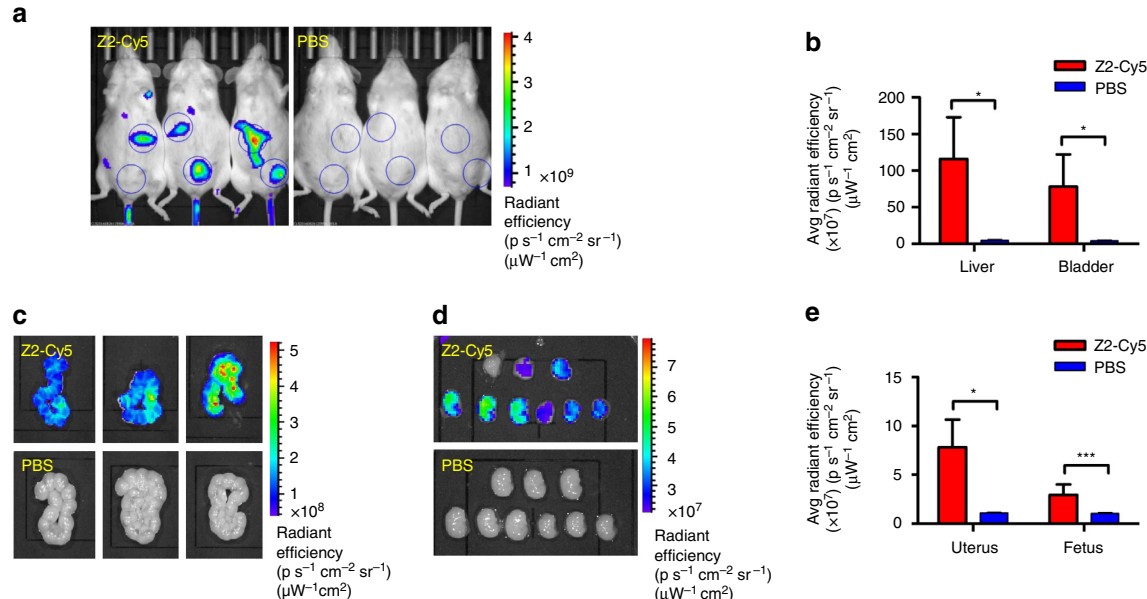

**Figure 4 | Ability of Z2 to penetrate the placental barrier of pregnant ICR mice.** (**a**) Imaging of pregnant ICR mice treated with Z2-Cy5 or PBS by the IVIS Lumina K Series III from PerkinElmer. Mice were injected intravenously with 100 μg Z2-Cy5 ($n = 3$) or PBS ($n = 3$) as control (for background fluorescence measurement), followed by imaging analysis. (**b**) The statistical analysis of results from Fig. 4a. (**c**) Imaging of the uteruses from pregnant mice. (**d**) Imaging of the fetuses ($n = 9$) from uteruses. (**e**) The statistical analysis of results from Fig. 4c,d. Data are means ± s.d., $*P < 0.05$; $***P < 0.001$, Student's two-tailed $t$-test.

mice (Fig. 4b). Then the uteruses of the pregnant mice and the fetuses in their uteruses were removed out for evaluating the distribution of Z2-Cy5. As speculated, the signal of Z2-Cy5 was seen in the uteruses (Fig. 4c) and the fetuses (Fig. 4d and Supplementary Fig. 9c). The average radiant efficiency in uteruses ($P = 0.0148$, Student's two-tailed $t$-test) and fetuses ($P < 0.0001$, Student's two-tailed $t$-test) from the Z2-Cy5 group was significantly higher than that from PBS group (Fig. 4e). In corroboration with the data in Supplementary Fig. 10, these results suggest that Z2 peptide injected intravenously or intraperitoneally (i.p.) can penetrate through the placental barrier of pregnant ICR mice and enter into the bodies of fetuses.

**Z2 is safe for pregnant ICR mice and their fetuses.** ZIKV usually causes mild symptoms, such as fever and rash, but when it infects pregnant women, it can cause congenital brain development abnormalities to fetus[6,38]. Therefore, the safety of an anti-ZIKV agent for pregnant women should be taken into consideration. Accordingly, we assessed the safety of Z2 in pregnant ICR mice. The results showed that body weight changes of mothers (Fig. 5a) and pups (Fig. 5b) were almost the same among the six groups (intravenous injection of Z2 at 10, 20, 40, 80 and $120 \, \mathrm{mg \, kg^{-1}}$, respectively, or PBS), suggesting that injection of Z2 does not cause visible damage to prenatal and postpartum health of the pregnant mice, nor does it interfere with the normal growth of pups. We did not find any pups with abnormal behaviour.

The levels of alanine aminotransferase (ALT; Fig. 5c) and creatinine (Fig. 5d) in the sera of mice in Z2- and PBS-treated groups showed no significant difference ($P > 0.05$, Mann–Whitney test) at all time points, suggesting that injection of Z2 at high or low doses does not affect the hepatic and renal function of pregnant mice. Using enzyme-linked immunosorbent assay, we measured Z2-specific antibodies in sera of pregnant mice at 1 or 2 weeks after the third injection of Z2 or PBS, and found that Z2-specific antibodies in both Z2- and PBS-treated

mice were at a level below detection (Supplementary Fig. 11). This result suggests that Z2 has poor immunogenicity.

We then compared the potential histopathological changes of mothers and pups among the six groups. As shown in Fig. 5e, the haematoxylin-and-eosin-stained sections of livers, kidneys, brains and spleens from mice treated with Z2 at different doses exhibited no pathological abnormality, when compared with those from mice treated by PBS. None of these samples showed evidence of cell degeneration, necrosis or infiltration of inflammatory factors. Overall, Z2 is safe for pregnant ICR mice and fetuses, even at the dose as high as $120 \, \mathrm{mg \, kg^{-1}}$ of body weight, which is 11-fold higher than that providing protection against ZIKV infection *in vivo*.

**Z2 blocked vertical transmission of ZIKV in pregnant mice.** To determine whether Z2 could protect against vertical transmission of ZIKV, pregnant C57BL/6 mice were infected by ZIKV as described previously[39] and were then treated with Z2 at $10 \, \mathrm{mg \, kg^{-1}}$ of body weight ($n = 12$) or vehicle control ($n = 12$). The results showed that Z2 treatment could reduce viraemia in ZIKV-infected pregnant C57BL/6 mice ($P = 0.0141$, Mann–Whitney test; Fig. 6a). At the same time, viral RNA load in placentas from Z2-treated pregnant mice was significantly lower than that from vehicle-treated mice ($P = 0.0029$, Mann–Whitney test), and the infection rate decreased from 18/24 to 12/24 (Fig. 6b). Interestingly, Z2 treatment resulted in the decrease of infection rate of fetal head from 14/24 to 2/24 ($P = 0.0001$, Mann–Whitney test; Fig. 6c). These results suggest that Z2 may inactivate ZIKV virions either before or after the virions have penetrated the placenta to fetus, thus reducing the infection rate of fetuses, as well as protecting against vertical transmission of ZIKV in pregnant mice.

**Z2 protected A129 or AG6 mice from lethal ZIKV challenge.** Finally, the antiviral efficacy of Z2 was confirmed in the recently established A129 (type I interferon receptor-deficient)[15] or AG6 (type I/II interferon receptor-deficient)[40,41] mouse model for

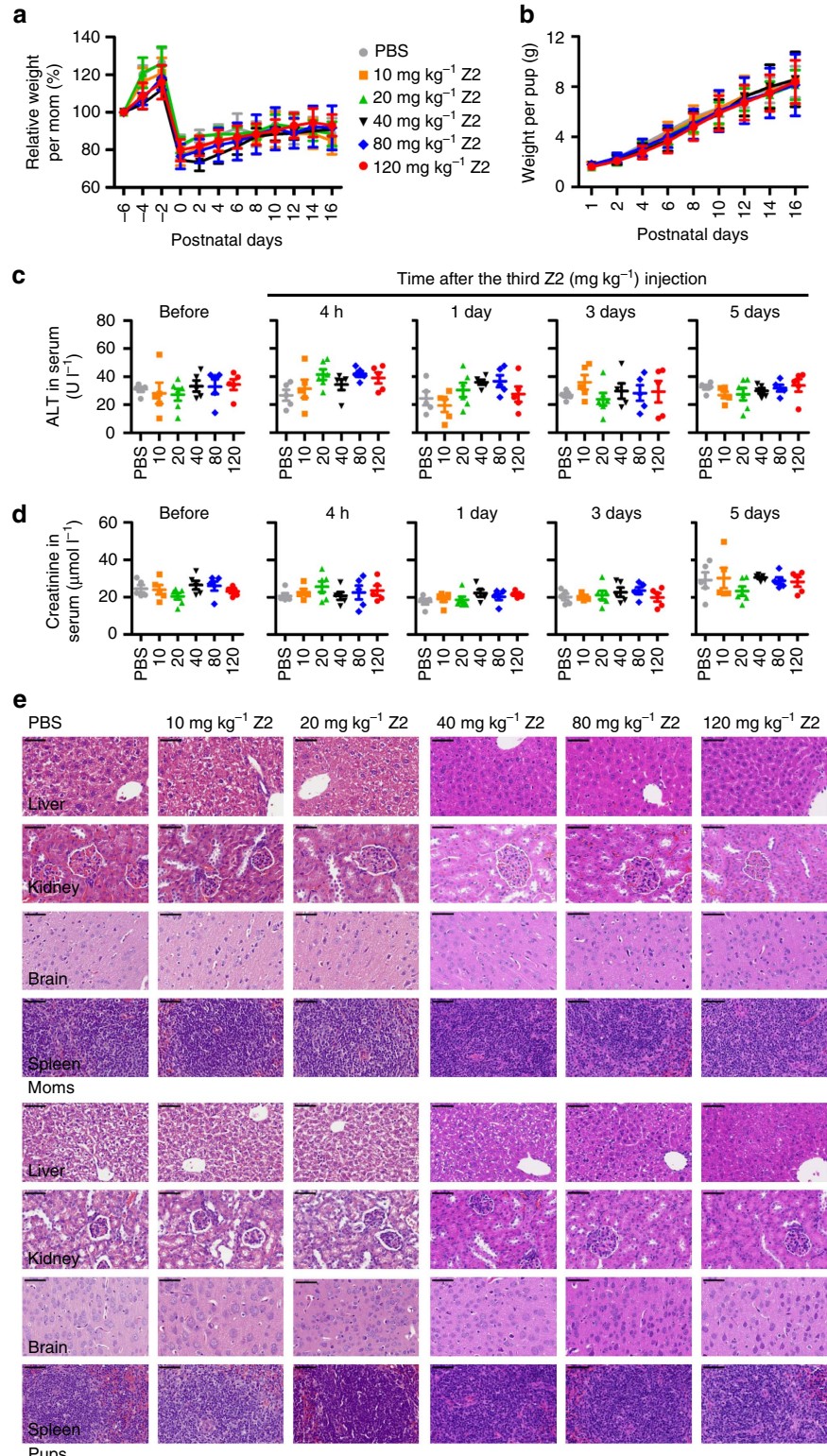

**Figure 5 | Safety analysis of Z2 for pregnant ICR mice and fetuses.** (**a**) Body weight changes of mothers at different prenatal and postnatal time points. Thirty-one pregnant ICR mice (E12–14) were assigned randomly to six groups and were injected intravenously with PBS ($n=5$), or PBS containing Z2 at escalating dose (10 mg kg$^{-1}$, $n=5$; 20 mg kg$^{-1}$, $n=6$; 40 mg kg$^{-1}$, $n=5$; 80 mg kg$^{-1}$, $n=5$; 120 mg kg$^{-1}$, $n=5$) every day for 3 consecutive days. Data are means ± s.d. (**b**) Body weight changes of pups at various postnatal time points. The average litter size of the offspring of PBS-treated pregnant mice was 10.4 ± 1.5, while that of the pregnant mice treated with Z2 at dose of 10, 20, 40, 80 and 120 mg kg$^{-1}$ was 10.2 ± 1.6, 10.5 ± 2.0, 10.4 ± 2.1, 10.6 ± 1.1 and 10.8 ± 2.6, respectively. The same legend was used for Fig. 5a,b. Data are means ± s.d. (**c**) The ALT in the sera of the pregnant mice measured by the ALT assay kit (NJJCBIO) before the first injection and 4 h, 1, 3 and 5 days after the third injection of Z2. All error bars reflect s.d. (**d**) The creatinine in the sera of the pregnant mice measured by the creatinine assay kit (NJJCBIO). All error bars reflect s.d. (**e**) Comparison of haematoxylin and eosin staining of tissues, including livers, kidneys, brains and spleens from mothers and pups in each group. Scale bar, 50 μm.

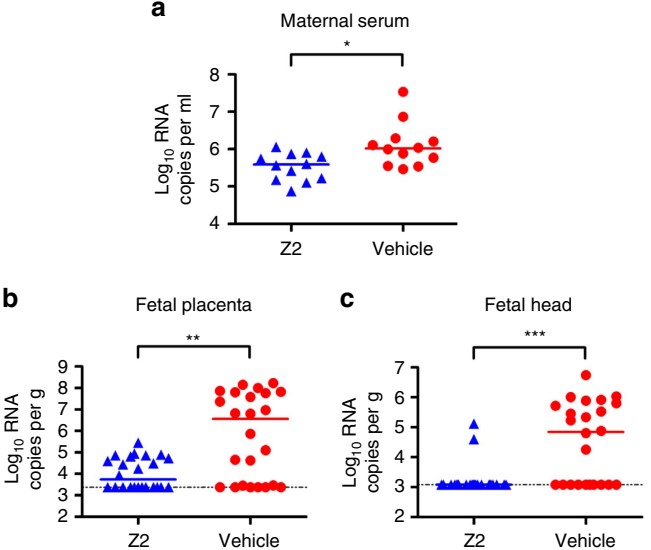

**Figure 6 | Protection against vertical transmission of ZIKV in Z2-treated pregnant C57BL/6 mice.** (**a**) Viraemia of pregnant C57BL/6 mice. Pregnant C57BL/6 mice were infected by ZIKV for 1h and treated with Z2 or vehicle as control. At day 1 post infection, sera were collected by retro-orbital bleeding for viraemia detection. (**b**) Viral RNA load in placentas. Two embryos of each pregnant mouse were randomly collected and the viral RNA load in each placenta was determined by RT–qPCR. (**c**) Viral RNA load in fetal heads. The viral RNA load in fetal head of each collected embryo was determined by RT–qPCR. All bars reflect median values, $*P < 0.05$; $**P < 0.01$; $***P < 0.001$, Mann–Whitney test.

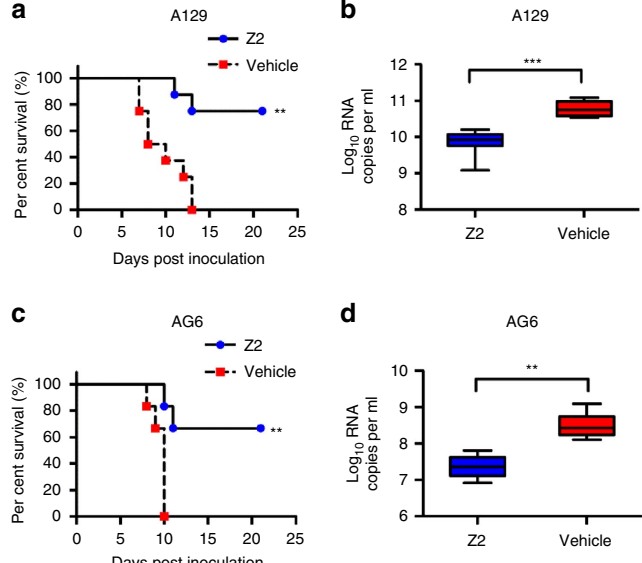

**Figure 7 | Protective activity of Z2 against ZIKV infection in lethal mouse models.** (**a**) Survival of ZIKV-infected A129 mice. A129 mice (4 weeks old) were infected with $1 \times 10^5$ p.f.u. of ZIKV through the intraperitoneal injection route. After 1h, mice were treated with Z2 ($n = 8$) at 10 mg kg$^{-1}$ of body weight, and vehicle ($n = 8$) as control. Mouse survival was observed and recorded daily until 21 d.p.i. $**P < 0.01$, log-rank (Mantel Cox) test. (**b**) Viral RNA load in sera of ZIKV-infected A129 mice. At day 2 post infection, mice were retro-orbitally bled to measure viral RNA load in sera by RT–qPCR. $***P < 0.001$, Mann–Whitney test. (**c**) Survival of ZIKV-infected AG6 mice. AG6 mice (6 weeks old) were infected with $1 \times 10^3$ p.f.u. of ZIKV via a subcutaneous route in the footpad. After 1h, mice were treated with Z2 ($n = 6$) at 10 mg kg$^{-1}$ of body weight, and vehicle ($n = 6$) as control. Mouse survival was observed daily and recorded until 21 d.p.i. $**P < 0.01$, log-rank (Mantel Cox) test. (**d**) Viral RNA load in sera of ZIKV-infected AG6 mice. At day 2 post infection, mice were retro-orbitally bled to measure viral RNA load in sera by RT–qPCR. Whiskers: 5–95 percentile. $**P < 0.01$, Mann–Whitney test.

ZIKV infection. One hour after the inoculation with ZIKV i.p., A129 mice were treated with Z2 or vehicle. As shown in Fig. 7a, all the mice treated with vehicle developed neurological symptoms from 5 days post inoculation and finally displayed a 100% mortality rate at 13 days post inoculation. In contrast, treatment with Z2 protected 75% of the A129 mice from death caused by ZIKV infection, and the survivors had no neurological symptoms. Z2 treatment also significantly prolonged mean survival time (MST) from 9 days to the end of the experiment ($P = 0.0010$, log-rank (Mantel Cox) test). The viral load in Z2-treated A129 mice at 2 days post infection (d.p.i.) was about 7-fold lower than that of vehicle-treated mice ($P = 0.0002$, Mann–Whitney test; Fig. 7b). Similarly, treatment with Z2 protected 67% of AG6 mice from death caused by subcutaneous administration of ZIKV and significantly prolonged MST from 10 days to the end of the experiment ($P = 0.0048$, log-rank (Mantel Cox) test; Fig. 7c). The viral load in Z2-treated AG6 mice at 2 d.p.i. was about 13-fold lower than that of vehicle-treated mice ($P = 0.0022$, Mann–Whitney test; Fig. 7d).

Twenty-four hours after inoculation with ZIKV, treatment with Z2 still could protect 33.3% of A129 mice from death ($P = 0.0139$, log-rank (Mantel Cox) test; Supplementary Fig. 12a). Viral load in the Z2-treated A129 mice at 3 d.p.i. was about 4-fold lower than that of the vehicle-treated mice ($P = 0.0043$, Mann–Whitney test; Supplementary Fig. 12b). Although Z2 inactivates ZIKV at the early stage of viral replication, consecutive Z2 injection after ZIKV penetration of cells could still provide some protection of the infected A129 mice, possibly by inactivating newly produced ZIKV virions and prevent their infection of more target cells.

## Discussion

It has been reported that peptides derived from the stem region of DENV E protein inhibit the infection of DENV type 1–4 and some other flaviviruses, including Russian spring summer encephalitis virus, Central European encephalitis virus and West Nile virus[27–29,42]. However, to the best of our knowledge, none of these peptides has been reported to be effective against ZIKV infection *in vitro* and *in vivo*, or be further developed in preclinical and clinical studies. Here we showed that peptide Z2 derived from the stem region of ZIKV E protein is highly effective in inhibiting *in vitro* infection of ZIKV, both in BHK21 and Vero cells, and other flaviviruses, such as DENV-2 and YFV 17D. Most importantly, intraperitoneal administration of Z2 protected pregnant C57BL/6 mice against vertical transmission of ZIKV and protected A129 or AG6 mice, which are very susceptible to ZIKV infection, against lethal ZIKV challenge. The mechanism by which a peptide derived from the stem region of a flavivirus can inhibit infection by a broad spectrum of flaviviruses remains a point of controversy. It was reported that DN59, a 33-mer peptide that mimics a fragment of stem region of DENV E protein, acts like a disrupter of the DENV membrane, possibly inducing hole formation, leading to release of the viral genome[28]. Similarly, we found that Z2 peptide could bind to the E protein of ZIKV and disrupt the integrity of ZIKV membrane, resulting in the inactivation of virions. However, no detailed molecular mechanism or signalling pathway has so far been elucidated that would explain how Z2 binds E protein and disrupts the viral cell membrane. Electrostatic and hydrophobic interactions[43,44] or viral capsid dynamic studies[45,46] may explain the possible

mechanism, but this investigation is beyond the scope of the present paper.

Interestingly, Z2 can disrupt flavivirus membranes, but has no effect on the integrity of pseudotyped VSV and MERS-CoV membranes and cell membranes, possibly because the lipid composition, protein components, charge and hydrophobicity among these membranes are different[43,44], especially for the flavivirus membranes, which are derived from internal endoplasmic reticulum membranes of infected cells. Nevertheless, we cannot exclude other putative mechanisms of action involved in the anti-ZIKV activity of Z2, such as inhibition of membrane fusion[27,29]. However, we do not have sufficient evidence to conclude the similar mechanism for Z2.

Because of the same vector, *Aedes aegypti*, cases of co-infections by ZIKV, DENV and other arboviruses, such as chikungunya virus[47,48] have been reported. Moreover, preexisting DENV antibodies may enhance ZIKV infection through antibody-dependent enhancement (ADE) effect, a phenomenon in which antibody for certain flavivirus enhances the infection by heterogeneous virus[49]. Similar to DENV, some ZIKV antibodies may also induce ADE effect on ZIKV and DENV infection[50,51]. Modification of the anti-ZIKV antibodies to decrease their binding to FcγR may reduce the risk of ADE effect[16]. However, this may further increase the cost of these antibodies. Z2, with antiviral activity against a broad spectrum of flaviviruses, may be favourable for patients infected by DENV and ZIKV simultaneously without the concern of ADE effect.

Z2, as a viral inactivator, has some advantages compared to most small-molecule antiviral drugs that target the viral replication stages inside the target cell, such as protease inhibitors. Z2 can inhibit ZIKV infection before viral entry and is, therefore, beneficial for treatment of ZIKV viraemia, especially for women of child-bearing age and their partners. The low toxicity *in vitro* and *in vivo* of Z2 ensures the safety of Z2 for pregnant women and their fetuses. Protease inhibitors, on the other hand, will not work until the virus enters into the cells, which, in turn, may allow more influx of virions to fetuses. Without the specificity against ZIKV, protease inhibitors may cause harmful effect to high-risk pregnant women. Therefore, combinational use of Z2 with a protease inhibitor for treatment of ZIKV infection may have synergistic effect, resulting in the reduction of dose, and thus the toxicity of the protease inhibitor.

Actually, ZIKV can infect pregnant woman and cause severe congenital brain developmental abnormalities in fetus, as well as can infect testis and cause damage that leads to male infertility[52,53]. Therefore, we also detected the distribution of Z2 in the genital organs of male mice. Supplementary Fig. 13 shows that Z2 could distribute to the genital organs of male mice, suggesting that it may prevent testis damage caused by ZIKV. However, further experiments need to be carried out to confirm this finding.

More versatile than traditional drugs, Z2 may be converted into bullets to target the virus in the fetal brain. Coupled with blood–brain barrier shuttle peptides, such as angiopep-2, glutathione, or peptides derived from transferrin (TfR1), or low-density lipoproteins (LDLRs)[54], Z2 may acquire the ability to penetrate through the fetal blood–brain barrier.

It should also be disclosed that peptides, as a drug, have some unfavourable features, such as immunogenicity and short half-life. Our study showed that intraperitoneal administration of Z2 peptide without an adjuvant did not induce anti-Z2 antibodies. The half-life time of Z2 is 2.767 h (Supplementary Fig. 14 and Supplementary Table 1). Nowadays, many ways have been found to prolong the half-life of peptide drugs, such as protecting N and C termini by N-acetylation and C-amidation, substituting L-amino acids with D-amino acids, and modification of amino acids with polyethylene glycol (PEG), albumin or lipopeptides[55,56].

In conclusion, peptide Z2 from the stem region of ZIKV envelope protein exhibits potent anti-ZIKV activity, and excellent safety and pharmacological profiles, indicating its potential for further development as a novel antiviral drug to treat ZIKV infection in high-risk populations, particularly the pregnant women.

## Methods

**Ethics statements and mice.** All animal experimental procedures were carried out according to ethical guidelines and approval by Institutional Laboratory Animal Care and Use Committee at Fudan University (20160927-2), Beijing Institute of Microbiology and Epidemiology (IACUC-13-2016-001), and Shanghai Public Health Clinical Center (2016-A021-01), respectively. The specific pathogen-free ICR pregnant mice and C57BL/6 pregnant mice at embryonic day 12–14 (E12–14) were purchased from Vital River Laboratories in Beijing and bred at the Department of Laboratory Animal Science of Fudan University. Specific pathogen-free A129 mice (female, 4 weeks of age) were bred at the Beijing Institute of Microbiology and Epidemiology. Specific pathogen-free AG6 mice (female, 6 weeks of age) were bred at the Animal Facility of Shanghai Public Health Clinical Center.

**Cells and viruses.** C6/36 (gift from Dr Yunwen Hu at Shanghai Public Health Clinical Center), BHK21 (American Type Culture Collection (ATCC), #CCL-10), Vero (ATCC, #CCL-81), 293T (ATCC, #CRL-3216) and Huh7 (Cell Bank of Chinese Academy of Science, #TCHu182) cells were cultured in Dulbecco's modified Eagle's medium (DMEM, Invitrogen, Carlsbad, CA) containing 10% fetal bovine serum (FBS, Biowest, France). ZIKV strain SZ01/2016 (GenBank number: KU866423) was isolated from a patient who returned from Samoa[57]. ZIKV strain GZ01/2016 (GenBank number: KU820898) was isolated from a patient who returned from Venezuela[58]. YFV strain 17D was obtained from Beijing Tiantan Biological products., Ltd and prepared as described previously[59]. ZIKV strains FLR (#VR1844) and MR766 (#VR1838) were obtained from ATCC. DENV-2 was kindly provided by Drs Yunwen Hu and Zhigang Song at the Shanghai Public Health Clinical Center. All the flaviviruses were propagated in C6/36 cells for the *in vitro* and *in vivo* experiments. The virus stock titre was calculated and diluted to 0.01 multiplicity of infection for *in vitro* experiments.

**Petides synthesis.** Peptides Z2(MAVLGDTAWDFGSVGGALNSLGKGIHQIFG AAF), Z2-Cy5 and the scrambled peptide of Z2 (Z2-scr, LDIIAGLSAGFQGGAT FVDAHGMVKASFLGGNW) were synthesized at GL Biochem (Shanghai, China) with 95% purity.

**Assays for antiviral activity.** Peptides (Z2 or Z2-scr) were dissolved in DMSO and serially diluted in serum-free DMEM. Then, 50 μl of virus (ZIKV, DENV-2 or YFV17D) were added to 50 μl peptide solution. After incubation at 37 °C for 1.5 h, the mixture was added to the cells seeded in 96-well plates, followed by incubation at 37 °C for 12 h. The culture supernatant was replaced with fresh DMEM containing 2% FBS. About 8 days later when ZIKV-induced CPE became evident, antiviral activity was detected using Cell Counting Kit-8 (CCK8, Dojindo, Japan) according to the instruction manual. Data were then collected by microplate reader (Infinite M200PRO, Tecan, USA).

The pseudotyped VSV and MERS-CoV were set as unrelated virus controls and prepared as previously described[60,61]. Briefly, 293T cells were co-transfected with a plasmid encoding MERS-CoV S protein provided by Dr Lanying Du at the New York Blood Center or VSV-G protein (gift from Dr Yan Wang at the Shanghai Institutes for Biological Sciences) and pNL4-3.luc.RE (obtained from NIH AIDS Reagent Program) using VigoFect (Vigorous Biotechnology, Beijing, China). Supernatants containing pseudotyped MERS-CoV or VSV were collected 48 h post transfection. Huh7 cells were infected by these control viruses as described above. After 3 days, the inhibitory activity of Z2 on pseudovirus infection was measured using the luciferase assay system according to the instruction manual (Promega Co., Madison, WI)[62].

**Immunofluorescence staining.** To determine the inhibitory activity of Z2 on ZIKV infection in BHK21 or Vero cells, cells ($1 \times 10^5$) were seeded onto coverslips in a 24-well plate the day before infection. Then, ZIKV was mixed with equal volume of Z2 serially diluted (20, 10 and 5 μM) or Z2-scr (20 μM) in serum-free DMEM. After incubation for 1.5 h at 37 °C, the mixture was added to cells. Twelve hours later, the culture supernatant was replaced with DMEM containing 2% FBS. After 4 days, the cells were fixed by 4% paraformaldehyde (PFA; Sigma-Aldrich, St Louis, MO), perforated by 0.2% Triton X-100 and blocked with 3% BSA (Amresco, LLC, Solon, OH). Then, the cells were incubated overnight with anti-E mAb 4G2 (10 μg ml⁻¹, provided by Drs Lifang Jiang and Danyun Fang at Sun Yat-sen University) at 4 °C. After five washes, the cells were incubated with Alexa Fluor 488-labelled donkey anti-mouse IgG (1:1,000, Thermo Fisher Scientific, Wilmington, DE, USA) at room temperature for 1 h. After five washes, the coverslips were sealed with Prolong Gold Antifade reagent with 4,6-diamidino-2-

phenylindole (Thermo Fisher Scientific) and scanned with the Leica SP8 confocal microscope.

To detect the expression of ZIKV E protein, 293T cells ($5 \times 10^4$) seeded on coverslips were transfected with pcDNA3.1-Env provided by Dr Lanying Du at the New York Blood Center, or empty vector pcDNA3.1 (Thermo Fisher Scientific) using VigoFect. After culture for 48 h, immunofluorescence staining was performed using anti-E mAb 4G2 as described above.

To test the binding of Z2 to ZIKV-infected BHK21 cells, BHK21 cells seeded in coverslips were ZIKV- or mock-infected, fixed with 4% PFA, perforated by 0.2% Triton X-100 and blocked with 3% BSA. The cells were then incubated with anti-E mAb 4G2. After five washes, the cells were incubated with Alexa Fluor 488-labelled donkey anti-mouse antibody and Z2-Cy5 at room temperature for 1 h. After another five washes, the coverslips were sealed for scanning with the Leica SP8 confocal microscope.

**Flow cytometry assay.** The 293T cells ($2 \times 10^5$) were seeded into a six-well plate. After culture at 37 °C for 16 h, the cells were transfected by pcDNA3.1-Env or pcDNA3.1 with VigoFect for 48 h. The cells were digested by EDTA buffer to obtain a single-cell suspension and then treated with 25 nM Z2-Cy5 for 30 min. The untreated cells were tested as the blank control. After three washes, the data of Z2-Cy5 binding to cells with or without E protein expression were collected with a BD Accuri C6 Flow Cytometer.

**RNase digestion assay and RT–qPCR.** The released genomic RNA from the ZIKV particles was detected by RNase digestion assay and RT–qPCR as previously described[28]. Briefly, about $1 \times 10^3$ plaque-forming units (p.f.u.) of ZIKV was incubated with Z2 or Z2-scr at room temperature for 2 h. The released genomic RNA from the treated ZIKV particles was then digested with micrococcal nuclease (New England BioLabs, MA) at 37 °C for 1 h. After inactivation of the residual RNase, the undigested genomic RNA in the intact viral particles was extracted using the Qiagen QIAamp Viral RNA Mini Kit (Valencia, CA) and reversed by using RT Reagent Kit (Takara Bio, Shiga, Japan). ZIKV RNA genome was quantified by SYBR PremixExTaqII (TliRNase H Plus from Takara Bio) and the Master Cycler Ep Realplex PCR System (Eppendorf, Hamburg, Germany) according to the manufacturers' instructions. The following primers were used to detect the RNA sequences in viral genome coding precursor membrane protein (PrM), E and capsid (Cap) proteins, respectively (Fig. 3c,d): PrM F1 (5′-CTTGGA CAGAAACGATGCTGGG-3′)/PrM R1 (5′-TGATGGCAGGTTCCGTACACA A-3′); E F1 (5′-TGGAGGCTGAGATGGATGG-3′)/E R1 (5′-GAACGCTGCGG TACACAAGGA-3′); and Cap F1 (5′-TCACGGCAATCAAGCCATCACT-3′)/Cap R1 (5′-GCCTCGTCTCTTCTTCTCCTT-3′).

**Sucrose density gradient assay.** Approximately $3 \times 10^5$ p.f.u. of ZIKV was treated with PBS containing 1% (v/v) DMSO, 100 μM Z2 or Z2-scr in PBS containing 1% (v/v) DMSO or PBS containing 1% (v/v) Triton X-100 at 37 °C for 2 h. Then the treated virions were gently loaded to the top of sucrose step gradient (20, 30, 40, 50, 60 and 70%) and then centrifuged in a swinging bucket rotor (SW41Ti, Beckman Coulter, Brea, CA) in an Optima L-100 XP ultracentrifuge (Beckman Coulter) for 3 h at 107,170$g$ at 4 °C. Fractions from top to bottom were collected and assayed for viral genomic RNA by RT–qPCR and E protein by western blot. Anti-E mAb 4G2 (10 μg ml$^{-1}$) was used to detect the E protein[50].

**Assay to detect inactivated ZIKV virions.** Inactivated ZIKV particles were separated and detected as previously described[36]. Briefly, 100 μl Z2 or Z2-scr at graded concentration (2, 10, 20, 50 and 100 μM) were added to 100 μl ZIKV ($5 \times 10^3$ p.f.u. per ml), followed by incubation at room temperature for 2 h. Then, 50% PEG-8000 (Amresco) and 5 M NaCl were added to the treated virus at final concentration of 10% and 0.67 M, respectively. After incubation on ice for 2 h, the mixture was centrifuged at 20,200$g$ for 1 h. The supernatant containing the free peptide was removed, and the pellet containing the viral particles was washed with 3% PEG-8000 in PBS containing 10 mg ml$^{-1}$ BSA (Amresco). After centrifugation, the pellet was resuspended in 200 μl DMEM containing 2% FBS. The infectivity of the ZIKV particles in the pellet was determined using BHK21 cells as described above.

***In vivo* fluorescence imaging.** Six pregnant ICR mice (10–12 weeks old, E12–14) were assigned randomly to two groups and injected intravenously with 100 μg Z2-Cy5 in 100 μl PBS ($n = 3$) or 100 μl PBS without peptide ($n = 3$) as control (for background fluorescence measurement). One hour later, mice were imaged for distribution of Z2-Cy5 by the IVIS lumina K series III *in vivo* imaging system (PerkinElmer, Waltham, MA) and the radiant efficiency (p s$^{-1}$ cm$^{-2}$ sr$^{-1}$) (μW$^{-1}$ cm$^2$) was calculated by Living image 4.4 software[63]. To determine the distribution of Z2-Cy5 in organs, all pregnant mice were killed by intraperitoneal injection of sodium pentobarbital and the dissected livers, kidneys, spleens, hearts, uteruses and fetuses were imaged and the average radiant efficiency was calculated. The fetuses in the uteruses of two groups were applied to PFA-fixed paraffin sections to demonstrate the distribution of Z2 in the placenta and fetus.

**Safety of Z2 for pregnant ICR mice and fetuses.** Thirty-one pregnant ICR mice (10–12 weeks old, E12–14) were assigned randomly to six groups and injected intravenously with Z2 at 10 ($n = 5$), 20 ($n = 6$), 40 ($n = 5$), 80 ($n = 5$) and 120 mg kg$^{-1}$ ($n = 5$) of body weight, or PBS ($n = 5$) as control every day for 3 consecutive days. Body weight changes of mothers at different prenatal and postnatal time points and pups at various postnatal time points were monitored. ALT and creatinine in the sera collected from the tails were measured using the ALT and creatinine assay kits (NJJCBIO, Nanjing, China) before the first injection and 4 h, 1, 3 and 5 days, respectively, after the third injection of Z2. Specific antibody response elicited by Z2 in mice was evaluated by enzyme-linked immunosorbent assay at 1 or 2 weeks after the third injection of Z2 or PBS. Three weeks after the pups were born, two mothers and their pups in each group were killed and the livers, kidneys, spleens and brains were collected for haematoxylin and eosin staining. All animal studies with ICR mice were carried out according to the ethical guidelines and approval by the Institutional Laboratory Animal Care and Use Committee at Fudan University (20160927-2).

**Antiviral efficacy of Z2 in pregnant C57BL/6 mice.** Antiviral efficacy of Z2 in pregnant mice was evaluated using the method as previously reported[39]. Briefly, 24 pregnant C57BL/6 mice (10–12 weeks old, E12–14) were assigned randomly to two groups and infected i.p. with $1 \times 10^5$ p.f.u. of ZIKV (SZ01). One hour later, the infected mice were i.p. administered with Z2 at 10 mg kg$^{-1}$ of body weight ($n = 12$) or vehicle control ($n = 12$). At day 1 post infection, mice were retro-orbitally bled to measure viraemia by RT–qPCR. Two embryos of each pregnant mouse were randomly collected and the viral RNA load in placenta and fetal head of each collected embryo was determined by RT–qPCR.

**Antiviral efficacy of Z2 in A129 or AG6 mice.** Antiviral efficacy of Z2 in an A129 or AG6 mouse model was evaluated as previously reported[15,40,41]. Briefly, sixteen 4-week-old female A129 mice were assigned randomly to two groups and infected i.p. with $1 \times 10^5$ p.f.u. of ZIKV (GZ01). Then, the infected mice were i.p. administered with Z2 at 10 mg kg$^{-1}$ of body weight ($n = 8$) or vehicle control ($n = 8$) every day for 6 consecutive days. Mice were observed daily for signs of illness and mortality. Animals that survived to day 21 were deemed to be protected. Meanwhile, viral RNA loads in sera on day 2 d.p.i. were measured by RT–qPCR. Twelve 6-week-old sex-matched AG6 mice were assigned randomly to two groups and infected with $1 \times 10^3$ p.f.u. of ZIKV (SZ01) via a subcutaneous route in the footpad. Then, the infected mice were i.p. administered with Z2 at 10 mg kg$^{-1}$ of body weight ($n = 6$) or vehicle control ($n = 6$) every day for 6 consecutive d.p.i. Then the observation and examination on AG6 mice were done as described above. All animal studies with infectious ZIKV were conducted in a Biosafety Level 2 facility at Beijing Institute of Microbiology and Epidemiology or Shanghai Public Health Clinical Center with Institutional Biosafety Committee approval.

**Statistical analysis.** Statistical methods were performed as follows: the log-rank (Mantel Cox) test to compare the MST between ZIKV-infected mice treated with Z2 or vehicle; Student's unpaired two-tailed $t$-test to monitor the distribution of Z2 in ICR mice; and the non-parametric Mann–Whitney test to examine the effect of Z2 on the levels of ALT and creatinine between the Z2 group and PBS group at one time point. The non-parametric Mann–Whitney test also to determine differences of viral RNA loads in the sera of mice treated with Z2 or vehicle. Statistical analyses were carried out by SPSS 13.0 and GraphPad Prism Software. *$P < 0.05$; **$P < 0.01$; ***$P < 0.001$.

**Data availability.** All relevant data are available from the authors on request.

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

## Acknowledgements

We thank Q. Wang at the sharing platform for large-scale instrument of Fudan University and all the staffs at the Animal Facility of Shanghai Public Health Clinical Center for their contribution to this study. This work was supported by grants from the Hi-Tech Research and Development Program of China (863 Program) (2015AA020930 to L.L.), the National Key Research and Development Program of China (2016YFC1201000 and 2016YFC1200405 to S.J., 2016YFD0500304 to C.-F.Q. and 2016YFC1202901 to L.L.), the National Natural Science Foundation of China (81630090 to S.J., 81522025 and 81661130162 to C.-F.Q., and 81672019 to L.L.), the Food and Health Bureau, the Government of the Hong Kong Special Administrative Region (ZIKA-HKU to J.F.-W.C.), and the Consultancy Service for Enhancing Laboratory Surveillance of Emerging Infectious Diseases of the Department of Health, Hong Kong Special Administrative Region (K.-Y.Y.).

## Author contributions

L.L., S.J. and C.-F.Q. designed the research, provided reagents and laboratory infrastructure; Y.Y., Y.-Q.D., P.Z., Q.W., Y.D., F.Y., L.D., N.-N.Z., M.T., J.-N.H., Y.M., Y.L., X.Z., J.F.-W.C. performed experiments and data analyses; Y.Y., L.L., C.-F.Q., J.F.-W.C., K.-Y.Y. and S.J. wrote the manuscript.

**Additional information**

**Competing interests:** The authors declare no competing financial interests.

