## [Peer Review File · Nature Communications]

Reviewers' comments:

Reviewer #1 (Remarks to the Author):

I read with great interest the manuscript entitled "A novel peptide inactivator of Zika virus with potential for the treatment of infected pregnant women" by Yu et al. This study provides a very promising discovery that a peptide, designated Z2, derived from the stem region of ZIKV E protein potently inhibits in vitro and in vivo infection of ZIKV. The authors have demonstrated that Z2 can interact with the ZIKV surface protein and disrupt the integrity of viral membranes, resulting in the release of viral genomic RNA and inactivation of ZIKV particles. Intraperitoneal administration of Z2 was highly effective in protecting type I interferon receptor-deficient mice against lethal ZIKV challenge. Most importantly, the authors have shown that Z2 could penetrate the placental barrier to enter fetus tissues and is safe for use in pregnant mice and their fetuses. Therefore, this peptide has great potential to be further developed as a safe and effective anti-ZIKV drug to treat ZIKV infection in high-risk populations, particularly the pregnant women.

Overall, the experiments in this study are performed rigorously and the results are of great importance. The paper is clearly written and will attract interest of researchers working in related areas and stimulate further progress in development of anti-ZIKV therapeutics. However, I have few recommendations that need to be considered in order to strengthen the manuscript and make the results and the claims of this paper more convincing for the scientific community. I had divided them into major and minor points.

Major points

1. Most recently, two papers were published to show that ZIKV could infect mouse testis and cause its damage, leading to male infertility (Nature. 2016 Oct 31. doi: 10.1038/nature20556; Cell. 2016, <http://dx.doi.org/10.1016/j.cell.2016.11.016>). The authors of this manuscript have demonstrated that Z2 peptide can be detected in most mouse organs, such as liver, kidney, spleen, heart, as well as placenta and fetal tissue. It would be very interesting if the authors are able to detect the presence of Z2 peptide in the mouse testis tissue. If Z2 can also enter the testis, application of this peptide may prevent the testis damage caused by ZIKV infection.
2. The authors have demonstrated that Z2 peptide can interact with ZIKV E protein expressed in the uninfected 293T cells. However, it is unclear whether Z2 can interact with the ZIKV surface protein (E/M) dimer . I would suggest the authors to show whether Z2 peptide can interact with the ZIKV surface protein expressed on the ZIKV-infected BHK21 cells.

Minor points

1. In Figure 1B, how were the amino acid residues in Z2 peptide numbered?
2. Please update the discussion of anti-ZIKV antibodies in lines 57 to 60 and 279 to 280 by citing the latest publication (Sapparapu, G. et al. Neutralizing human antibodies prevent

Zika virus replication and fetal disease in mice. Nature
<http://dx.doi.org/10.1038/nature20564> (2016)).

3. In line 71, when the abbreviation "E protein" first appears in the manuscript, it should be explained.
4. The English grammar needs some brush up, such as "the placenta barrier" (line 177), "such as fever, rash, and but " (line 193), etc.
5. In lines 184 to 187, Figure 4D appears before Figure 4B and 4C. This should be adjusted so that the figures appear in sequence.
6. In line 316, the subtitle in the Method section: "Assays for antiviral activity and cytotoxicity" should be changed to "Assays for antiviral activity". The cytotoxicity experiments were not included in this section.
7. In line 326, both reference 31 and 53 are related with pseudotyped MERS-CoV, but no reference was provided about the preparation of pseudo typed VSV.

Reviewer #2 (Remarks to the Author):

The authors describe the development and characterization of a peptide derived from ZIKV E protein to be used as a fusion inhibitor of ZIKV to be used in pregnant women. The manuscript details Z2's inhibitory properties against ZIKV in vitro, its distribution in vivo, and its ability to interrupt pathogenesis in a mouse model. The paper offers several novelties, including a peptide that could potentially move forward for use in humans, but the manuscript falls short, especially given a title that suggests that it may be beneficial for use in pregnant women. The paper does not even demonstrate this in mice, thus it is a stretch, and misleading, to write this in the title, as it is not a major finding.

Critiques, both major and minor:

1. Authors show that Z2 has protective activity in vivo, only when used 1 hour post infection. have the authors tried any other time? Is the drug capable of blocking transmission of ZIKV trans placentally? Without this data, this paper does not offer any potential for use in pregnant women as the title implies.
2. fig 2. any reason why a standard plaque assay was not used? Fig 2C, where is the control peptide as a control image?
Same critique for D. Scrambled peptide derived from Z2 would have been the best control to use throughout.
3. The flow plots do not look that different if the gates were drawn NOT to include half of what appears to be a negative population (minor shift in the population. probably the cells that are fluorescing around 105 on the X axis is positive, not the ones that are next to the negative population. Given that, there is a lot of background. Also, the authors should demonstrate that the cells that are ZIKV E-positive are the same ones where Z2 is binding. This should be fairly easy given that Z2 is already labeled with Cy5. Fig 3C, no control peptide control was used.
4. fig 4. how long after injection were mice imaged? what is the half-life of the peptide in

vivo?

5. not sure why figure 5A exists. a figure legend doesn't need its own panel in a figure. is there any information as to the litter size of the offspring or the total number of females that got pregnant in the PBS versus the Z2 receiving mice?

Reviewer #3 (Remarks to the Author):

Summary:

The authors present a new peptide drug (Z2 with 33 amino acids) with efficacy against Zika virus (ZIKV), which places a particular threat on pregnant women and, more specifically, on the developing fetus via teratogenic effects. The synthetic peptide Z2 is derived from the stem region of ZIKV envelope protein and is able to penetrate the placental barrier of pregnant, non-infected mice. Conversely, daily injection of Z2 protected non-pregnant mice from a lethal infection challenge with ZIKV. As other peptide drugs were previously used in pregnant women to prevent HIV transmission to the child and based on the presented data, this concept seems promising. As mode of action, the manuscript shows that Z2 disrupts the membrane of the enveloped ZIKV, leads to the release of viral genomes, and thus abrogates infectivity. The used methods are well suited to address the raised research questions, and the experiments are well controlled, including unrelated peptides or viruses. Nevertheless, several key points on the route of virus challenge and safety profile need to be addressed as outlined below before the manuscript may be suited for publication in Nature Communications.

MAJOR COMMENTS

1. The results that Z2 reaches fetuses of non-infected A129 mice (IVIS imaging of fluorescently labeled Z2, Fig. 4) seem disconnected from the data, showing that Z2 can protect adult not pregnant mice from ZIKV infection (survival data, Fig. 6). A major claim of the manuscript is that Z2 can be used during pregnancy to protect fetuses from ZIKV infection, but this claim is actually not addressed by the data. The authors need to provide evidence that Z2 can prevent teratogenic effects in pregnant A129 mice by protecting the developing fetuses from ZIKV infection. These experiments could be performed using a lower and sublethal dose of ZIKV in pregnant mice and determine fetal development.

2. Fig. 5: The authors claim that Z2 is a safe peptide drug candidate for use in pregnant women. Safety of Z2 was, however, only tested in pregnant mice the dose that protected non-pregnant mice from ZIKV infection (10 mg/kg) and a 2-fold higher dose (20 mg/kg). Even though not showing in vitro cytotoxicity for cell lines (Fig. S4) or mouse red blood cells (Fig. S4), the performed test in mice is not sufficient to conclude about toxicity in vivo. Along the same lines, the shift in fluorescence of E-protein negative cells in Fig. 3B may indicate a non-specific binding of Z2 to non-infected cells, and Z2 reaches non-infected fetuses (Fig. 4), suggesting binding to non-infected cells and potential off-target effects. Consequently, a dose-escalation study in pregnant mice needs to be performed to determine potential toxicity.

- To test the efficacy of Z2 *in vivo*, mice were injected intraperitoneally with Z2 one hour after infection with ZIKV with the same route. In addition, *in vitro* experiments were performed via pre-incubating ZIKV and Z2 for 1.5 hours. These procedures do not reflect the natural events that occur during ZIKV transmission via infected mosquitoes. The manuscript would largely benefit from showing that virus inoculation and treatment is independent. The authors could for instance inoculate ZIKV intravenous, subcutaneous, or intradermal to model natural routes of virus transmission and then inject Z2 intraperitoneal.

- Along the same line as the point above, testing the spread of Z2 to the fetus was performed via intravenous injection (line 677), but testing the efficacy of the drug against ZIKV infection was performed via *i.p.* injection. For testing the spread of Z2 to the fetus, the same route of inoculation should be performed as for testing the efficacy of the drug.

MINOR COMMENTS

- Line 46-47: The authors should mention the Americas in addition to Africa and Asia-Pacific region as areas at risk for ZIKV transmission.

Reviewers' comments:

Reviewer #1 (Remarks to the Author):

I read with great interest the manuscript entitled “A novel peptide inactivator of Zika virus with potential for the treatment of infected pregnant women” by Yu et al. This study provides a very promising discovery that a peptide, designated Z2, derived from the stem region of ZIKV E protein potently inhibits in vitro and in vivo infection of ZIKV. The authors have demonstrated that Z2 can interact with the ZIKV surface protein and disrupt the integrity of viral membranes, resulting in the release of viral genomic RNA and inactivation of ZIKV particles. Intraperitoneal administration of Z2 was highly effective in protecting type I interferon receptor-deficient mice against lethal ZIKV challenge. Most importantly, the authors have shown that Z2 could penetrate the placental barrier to enter fetus tissues and is safe for use in pregnant mice and their fetuses. Therefore, this peptide has great potential to be further developed as a safe and effective anti-ZIKV drug to treat ZIKV infection in high-risk populations, particularly the pregnant women.

Overall, the experiments in this study are performed rigorously and the results are of great importance. The paper is clearly written and will attract interest of researchers working in related areas and stimulate further progress in development of anti-ZIKV therapeutics. However, I have few recommendations that need to be considered in order to strengthen the manuscript and make the results and the claims of this paper more convincing for the scientific community. I had divided them into major and minor points.

Response: We thank the reviewer for the encouraging comments and helpful suggestions.

Major points

1. Most recently, two papers were published to show that ZIKV could infect mouse testis and cause its damage, leading to male infertility (Nature. 2016 Oct 31. doi: 10.1038/nature20556; Cell. 2016, <http://dx.doi.org/10.1016/j.cell.2016.11.016>). The authors of this manuscript have demonstrated that Z2 peptide can be detected in most mouse organs, such as liver, kidney, spleen, heart, as well as placenta and fetal tissue. It would be very interesting if the authors are able to detect the presence of Z2 peptide in the mouse testis tissue. If Z2 can also enter the testis, application of this peptide may prevent the testis damage caused by ZIKV infection.

Response: We have tested the distribution of Z2 in the testis and the other genital organs of male mice, such as seminal vesicle and epididymis. The results showed that Z2 could distribute to the genital organs of male mice (**Supplementary Figure 12**).

Supplementary Figure 12: Distribution of Z2 in the genital organs of male ICR mice. (A) Imaging of male ICR mice treated with Z2-Cy5 or PBS by the IVIS® Lumina K Series III from PerkinElmer. Mice were injected intravenously with 100 μg Z2-Cy5 (n=3) or PBS (n=3) as control (for background fluorescence measurement), followed by imaging analysis. (B) Imaging of the testis, seminal vesicle and epididymis from the male mice. (C) The statistical analysis of results from (B). Data are means \pm SD. ***, $p < 0.001$.

The related results were added in the revised manuscript (lines 308 - 313):

“Actually, ZIKV can infect pregnant woman and cause severe congenital brain developmental abnormalities in fetus, as well as can infect testis and cause damage that leads to male infertility^{52, 53}. Therefore, we also detected the distribution of Z2 in the genital organs of male mice. **Supplementary Figure 12** shows that Z2 could distribute to the genital organs of male mice, suggesting that it may prevent testis damage caused by ZIKV. However, further experiments need to be carried out to confirm this finding.”

2. The authors have demonstrated that Z2 peptide can interact with ZIKV E protein expressed in the uninfected 293T cells. However, it is unclear whether Z2 can interact with the ZIKV surface protein (E/M) dimer. I would suggest the authors to show whether Z2 peptide can interact with the ZIKV surface protein expressed on the ZIKV-infected BHK21 cells.

Response: We thank the reviewer for the helpful suggestion. We have determined the interaction between Z2 and ZIKV surface protein expressed on the ZIKV-infected

BHK21 cells. As shown in **Supplementary Figure 4**, E protein (green, stained by anti-E mAb 4G2) and Z2-Cy5 (red) overlapped and colocalized together (**Supplementary Figure 4A**). In the flow cytometry assay (**Supplementary Figure 4B**), Z2-Cy5 bound with 63.5% of ZIKV-infected BHK21 cells, significantly higher than that of mock-infected cells. The results suggest that Z2 peptide can interact with ZIKV surface protein expressed on ZIKV-infected BHK21 cells.

Supplementary Figure 4: Binding of Z2-Cy5 to ZIKV-infected BHK21 cells. (A) Immunofluorescence staining assay. Green, ZIKV E protein; Red, Z2-Cy5; Blue, nuclei. Scale bar = 100 μm. **(B)** Flow cytometry assay. After the incubation of ZIKV- or mock-infected BHK21 cells with Z2-Cy5, the cells were washed five times and processed by a BD Accuri C6 Flow Cytometer.

The result was added to the revised manuscript (**lines 144-145**): “Similar results were obtained from ZIKV-infected BHK21 cells (**Supplementary Figure 4**).”

The related methods were added in the revised manuscript (**lines 387 - 392**):

“To test the binding of Z2 to ZIKV- infected BHK21 cells, BHK21 cells seeded in cover slips were ZIKV or mock infected, fixed with 4 % PFA, perforated by 0.2% Triton X-100 and blocked with 3% BSA. The cells were then incubated with anti-E mAb 4G2. After 5 washes, the cells were incubated with Alexa Fluor 488-labeled donkey anti-mouse antibody and Z2-Cy5 at RT for 1 h. After another 5 washes, the cover slips were sealed for scanning with the Leica SP8 confocal microscope. ”

Minor points

1. In Figure 1B, how were the amino acid residues in Z2 peptide numbered?

Response: Z2 peptide was derived from the membrane-proximal stem region (residues 421-453) of ZIKV E protein. Therefore, we numbered the peptide as 421-453 in Figure 1B.

2. Please update the discussion of anti-ZIKV antibodies in lines 57 to 60 and 279 to 280 by citing the latest publication (Sapparapu, G. et al. Neutralizing human antibodies prevent Zika virus replication and fetal disease in mice. Nature <http://dx.doi.org/10.1038/nature20564> (2016)).

Response: Thanks for the suggestion. We have updated the discussion of anti-ZIKV antibodies in lines 57 to 60 and 279 to 280 in the original manuscript and cited the latest publication (Sapparapu, G. et al. Neutralizing human antibodies prevent Zika virus replication and fetal disease in mice. Nature <http://dx.doi.org/10.1038/nature20564> (2016)). We have added the following paragraphs:

“ZIKV-117, a human monoclonal antibody, could broadly neutralize infection of divergent ZIKV strains¹⁶. However, the high cost may limit its application in developing countries, such as Brazil.” (lines 58 - 60 in the revised manuscript).

“Modification of the anti-ZIKV antibodies to decrease their binding to FcγR may reduce risk of ADE¹⁶. However, this may further increase the cost of these antibodies.” (lines 293 - 295 in the revised manuscript).

3. In line 71, when the abbreviation “E protein” first appears in the manuscript, it should be explained.

Response: The abbreviation of “E protein” was explained upon its first appears in the manuscript.

4. The English grammar needs some brush up, such as “the placenta barrier” (line 177), “such as fever, rash, and but ” (line 193), etc.

Response: Thanks, we have done our best to correct the English grammar errors in the manuscript.

5. In lines 184 to 187, Figure 4D appears before Figure 4B and 4C. This should be adjusted so that the figures appear in sequence.

Response: The order of panels in **Figure 4** was adjusted accordingly.

6. *In line 316, the subtitle in the Method section: "Assays for antiviral activity and cytotoxicity" should be changed to "Assays for antiviral activity". The cytotoxicity experiments were not included in this section.*

Response: We have changed the subtitle according to the reviewer's suggestion.

7. *In line 326, both reference 31 and 53 are related with pseudotyped MERS-CoV, but no reference was provided about the preparation of pseudotyped VSV.*

Response: We apologize for this mistake. We have added the following paragraph: "The pseudotyped VSV and MERS-CoV were set as unrelated virus controls and prepared as previously described^{60,61}. Briefly, 293T cells were cotransfected with a plasmid encoding MERS-CoV S protein or VSV-G protein and pNL4-3.luc.RE using VigoFect. Supernatants containing pseudotyped MERS-CoV or VSV were harvested 48 h post-transfection. Huh7 cells were infected by these pseudotyped viruses as described above." (**lines 364 - 366** in the revised manuscript).

Reviewer #2 (Remarks to the Author):

The authors describe the development and characterization of a peptide derived from ZIKV E protein to be used as a fusion inhibitor of ZIKV to be used in pregnant women. The manuscript details Z2's inhibitory properties against ZIKV in vitro, its distribution in vivo, and its ability to interrupt pathogenesis in a mouse model. The paper offers several novelties, including a peptide that could potentially move forward for use in humans, but the manuscript falls short, especially given a title that suggests that it may be beneficial for use in pregnant women. The paper does not even demonstrate this in mice, thus it is a stretch, and misleading, to write this in the title, as it is not a major finding.

1. *Authors show that Z2 has protective activity in vivo, only when used 1 hour post infection. have the authors tried any other time? Is the drug capable of blocking transmission of ZIKV transplacentally? Without this data, this paper does not offer any potential for use in pregnant women as the title implies.*

Response: We thank the reviewer for the insightful comments, which help us to improve our manuscript. We changed the title to "**A novel antiviral peptide inactivates Zika virus and prevents its infection in pregnant mice and their fetuses**", and responded to the reviewer's critiques one by one as shown below.

1.1 Authors show that Z2 has protective activity *in vivo*, only when used 1 hour post infection. have the authors tried any other time?

Response: We did not try any other time study before. According to the constructive suggestion, we administered Z2 peptide to A129 mice 24 h after inoculation with ZIKV and found that about 33.3 % of these treated mice had survived (**Supplementary Figure 11A**). Viral load in the Z2-treated A129 mice at 3 days post-infection was about 4-fold lower than that of vehicle-treated mice ($p < 0.01$) (**Supplementary Figure 11B**). These results indicate that Z2 peptide still has protective activity *in vivo* even when administered 24 h after ZIKV infection.

Supplementary Figure 11: Protective activity of Z2 against ZIKV in A129 mice. (A) Survival of ZIKV-infected A129 mice. A129 mice (4 weeks old) were infected with 1×10^5 PFU of ZIKV through the intraperitoneal injection (i.p.) route. After 24 h, mice were treated with Z2 (n=6) at 10 mg/kg of body weight and vehicle (n=5) as control. Mouse survival was observed and recorded daily until 21 days post-infection. **(B)** Viral RNA load in sera of ZIKV-infected A129 mice. At day 3 post-infection, mice were retro-orbitally bled to measure viral RNA load in sera by RT-qPCR. *, $p < 0.05$; **, $p < 0.01$.

We have added the results in the revised manuscript:

“Twenty-four hours after inoculation with ZIKV, treatment with Z2 still could protect 33.3 % of A129 mice from death ($p < 0.05$) (**Supplementary Figure 11A**). Viral load in the Z2-treated A129 mice at 3 days post-infection was about 4-fold lower than that of the vehicle-treated mice ($p < 0.01$) (**Supplementary Figure 11B**). Although Z2 inactivates ZIKV at the early stage of viral replication, consecutive Z2 injection after ZIKV penetrance of cells could still provide some protection of the infected A129 mice, possibly by inactivating newly produced ZIKV virions and prevent their infection of more target cells.” (lines 251 - 257 in the revised manuscript)

During this study, we could not obtain more A129 mice from the animal suppliers in China; therefore, we will test more time points after administration of Z2 peptide in the future once we have obtained enough A129 mice for the experiments.

1.2 Is the drug capable of blocking transmission of ZIKV transplacentally?

Response: Yes, Z2 peptide could block transmission of ZIKV transplacentally in mice. We have added experiments to prove it (new **Figure 6**), and the related results in the revised manuscript (**lines 223 - 234**):

“To determine whether Z2 could protect against vertical transmission of ZIKV, pregnant C57BL/6 mice were infected by ZIKV as described previously³⁹ and were then treated with Z2 at 10 mg/kg of body weight (n=12) or vehicle control (n=12). The results showed that Z2 treatment could reduce viremia in ZIKV-infected pregnant C57BL/6 mice ($p < 0.05$) (**Figure 6A**). At the same time, viral RNA load in placentas from the Z2-treated pregnant mice was significantly lower than that from vehicle-treated mice ($p < 0.01$), and the infection rate decreased from 18/24 to 12/24 (**Figure 6B**). Interestingly, Z2 treatment resulted in the decrease of infection rate of fetal head from 14/24 to 2/24 ($p < 0.001$) (**Figure 6C**). These results suggest that Z2 may inactivate ZIKV virions either before or after the virions have penetrated the placenta to fetus, thus reducing the infection rate of fetuses, as well as protecting against vertical transmission of ZIKV in pregnant mice.”

Figure 6. Protection against vertical transmission of ZIKV in Z2-treated pregnant C57BL/6 mice. (A) Viremia of pregnant C57BL/6 mice. Pregnant C57BL/6 mice were infected by ZIKV for 1 h and treated with Z2 or vehicle control. At day 1 post-infection, sera were collected by retro-orbital bleeding for viremia detection. (B) Viral RNA load in placentas. Two embryos of each pregnant mouse were randomly collected and the viral RNA load in each placenta was determined by RT-qPCR. (C) Viral RNA load in fetal heads. The viral RNA load in fetal head of each collected embryo was determined by RT-qPCR. *, $p < 0.05$; **, $p < 0.01$; ***, $p < 0.001$.

The related methods were added in the revised manuscript (lines 456 - 463):

“Antiviral efficacy of Z2 in pregnant C57BL/6 mice

Antiviral efficacy of Z2 in pregnant mice was evaluated using the method as previously reported³⁹. Briefly, 24 pregnant C57BL/6 mice (E12-14) were assigned randomly to two groups and infected intraperitoneally (i.p.) with 1×10^5 PFU of ZIKV (SZ01). One h later, the infected mice were i.p. administered with Z2 at 10 mg/kg of body weight (n=12) or vehicle control (n=12). At day 1 post-infection, mice were retro-orbitally bled to measure viremia by RT-qPCR. Two embryos of each pregnant mouse were randomly collected and the viral RNA load in placenta and fetal head of each collected embryo was determined by RT-qPCR.”

2. fig 2. any reason why a standard plaque assay was not used? Fig 2C, where is the control peptide as a control image? Same critique for D. Scrambled peptide derived from Z2 would have been the best control to use throughout.

2.1 fig 2. any reason why a standard plaque assay was not used?

Response: Based on our previous experience¹ and others' publication², we quickly developed a colorimetric viral infection assay using CCK8 kit for screening anti-ZIKV compounds. Since we found this method to be convenient, quantitative, and reproducible, we used it to evaluate the anti-ZIKV activity of Z2 peptide. The validation of this colorimetric assay was recently confirmed by Muller et al³. To address your concerns, we performed the standard plaque assay to determine the anti-ZIKV activity of Z2 using BHK21 cells. As shown in **Supplementary Figure 2**, the result derived from the plaque assay (IC_{50} : $2.61 \pm 0.46 \mu M$) is consistent with that obtained from the CCK8 assay (IC_{50} : $1.75 \pm 0.13 \mu M$).

Supplementary Figure 2: Plaque reduction assay to determine ZIKV infection in BHK21 cells. (A) Plaque reduction assay for Z2. ZIKV was incubated with Z2 or Z2-scr in different concentrations for 1.5 h. Then the mixture was added to BHK21 cells seeded in 6-well plates. After 1.5 h, the viral inoculum was removed and 2.5 ml DMEM with 2% FBS and 1% low melting-point agarose overlaid the cells. Four to five days later, the cells were fixed with 4 % PFA and stained with 1% crystal violet overnight. (B) Plaque reduction curves. Plaques were counted and percentage of plaque reduction was calculated.

We added the results in the revised manuscript (**lines 111 - 113**): “We also used the plaque reduction assay and BHK 21 cells to test anti-ZIKV activity. As shown in **Supplementary Figure 2**, the IC_{50} is $2.61 \pm 0.46 \mu M$, suggesting that the result derived from the plaque reduction assay is consistent with that obtained from colorimetric CCK8 assay.”

2.2 Fig 2C, where is the control peptide as a control image?

Response: The control image of the control peptide was added to **new Figure 2C**, as shown below.

2.3 Same critique for D. Scrambled peptide derived from Z2 would have been the best control to use throughout.

Response: We agree. We designed the scrambled peptide of Z2 (Z2-scr: LDIIAGLSAGFQGGATFVDAHGMVKASFLGGNW) by using the Pep Controls (scramble pep) Program v.1.2 and synthesized this peptide. Using this scrambled peptide as a control, we have repeated a series of experiments to compare its anti-ZIKV activity with that of Z2 peptide. The new results were added to the following figures: **Figure 2A, 2B, 2C, 2D, 2E, 2F, 3C, 3D, 3E and 3F**. Based on all new data, Z2-scr showed no significant anti-ZIKV activity, while Z2 in these new experiments exhibited anti-ZIKV activity similar to that shown before in the original manuscript.

Original Figure 2

New Figure 2

Original Figure 3

New Figure 3

The related methods or results were added in the **Methods** and **Results** (highlighted in yellow) in the revised manuscript.

3. The flow plots do not look that different if the gates were drawn NOT to include half of what appears to be a negative population (minor shift in the population. probably the cells that are fluorescing around 10^5 on the X axis is positive, not the ones that are next to the negative population. Given that, there is a lot of background. Also, the authors should demonstrate that the cells that are ZIKV E-positive are the same ones where Z2 is binding. This should be fairly easy given that Z2 is already labeled with Cy5. Fig 3C, no control peptide control was used.

3.1 The flow plots do not look that different if the gates were drawn NOT to include half of what appears to be a negative population (minor shift in the population. probably the cells that are fluorescing around 10^5 on the X axis is positive, not the ones that are next to the negative population. Given that, there is a lot of background.

Response: We have drawn the gate to the cells that are fluorescing around 10^5 on the X axis. Indeed, the difference was reduced. The shedding or the residues of Cy5 from Z2-Cy5 may have contributed to the background.

3.2 Also, the authors should demonstrate that the cells that are ZIKV E-positive are the same ones where Z2 is binding. This should be fairly easy given that Z2 is already labeled with Cy5.

Response: As suggested by Reviewer #1, we have carried out immunofluorescence staining experiments using ZIKV-infected BHK21 cells with high expression level of E protein. We have demonstrated that the ZIKV E-positive cells are the same as those binding with Z2. Although we washed the cells for a longer time, some background staining remained. As shown in **Supplementary Figure 4**, the green (E protein, stained by 4G2) and red (Z2-Cy5) fluorescence overlapped, suggesting that Z2-Cy5 could bind to E protein in ZIKV-infected BHK21 cells.

Supplementary Figure 4: Binding of Z2-Cy5 to ZIKV-infected BHK21 cells. (A) Immunofluorescence staining assay. Green, ZIKV E protein ; Red, Z2-Cy5; Blue, nuclei. Scale bar = 100 μ m.

3.3 Fig 3C, no control peptide control was used.

Response: The scrambled peptide of Z2 (Z2-scr) was used as a control peptide and the new result was added to new **Figure 3C** in the revised manuscript.

4. fig 4. how long after injection were mice imaged? what is the halflife of the peptide *in vivo*?

Response: Mice were imaged 1 h after injection. The half-life of the peptide *in vivo* was 2.767 h, which was determined in SD rats after administration of Z2 by an AB SCIEX QTRAP 6500 instrument (LC-MS/MS, SCIEX, Boston, USA) in Multiple Reaction Monitoring mode ⁴. The new data were added to **Supplementary Figure 13** and **Supplementary Table 1**.

Supplementary Figure 13: Pharmacokinetic study of Z2. SD rats (200 ± 10 g, $n=3$) were administrated 10 mg/kg Z2 intravenously via the tail vein. Blood samples were collected through retro-orbital bleeding at 0.25, 0.5, 1, 2, 3, 4, 8 and 12 h after administration and centrifuged at 6000 rpm for 5 min to obtain the serum samples. Then 150 μ l methanol containing 15 μ g C24M peptide (MTWEEWDKKIEEYTKKIEELIKKS) as an internal standard was added to 50 μ l serum to precipitate the protein. After centrifugation at 17,000 rpm for 10 min, the supernatant was subjected to liquid chromatography-tandem mass spectrometry (LC-MS/MS) analysis, using an AB SCIEX QTRAP 6500 instrument (SCIEX, Boston, USA). All concentration data were dose-normalized and plotted as serum drug concentration time curves.

Supplementary Table 1: Pharmacokinetic parameters of Z2 in SD rats.

Parameter (unit)	Value
K_{el} (1/h)	0.250
$T_{1/2}$ (h)	2.767
T_{max} (h)	0.25
C_{max} (ng/ml)	754.6
$AUC_{(0-12\text{ h})}$ (ng/ml*h)	547.9
AUC_{inf} (ng/ml*h)	556.7

PK solutions 2.0 (noncompartmental pharmacokinetics data analysis) was utilized to analyze the pharmacokinetic parameters. K_{el} , elimination constant; $T_{1/2}$, half-life; T_{max} , the time to reach peak drug concentration in serum; C_{max} , peak drug concentration in serum; AUC, Area under the concentration-time curve.

The result was added to the revised manuscript (**lines 320-321**): “The half-life time of Z2 is 2.767 h (**Supplementary Figure 13 and Supplementary Table 1**).”

5. not sure why figure 5A exists. a figure legend doesn't need its own panel in a figure. is there any information as to the litter size of the offspring or the total number of females that got pregnant in the PBS versus the Z2 receiving mice?

5.1. not sure why figure 5A exists. a figure legend doesn't need its own panel in a figure.

Response: We apologize for the confusion. In our revised manuscript, we removed Figure 5A and added the following statement “The same legend was used for Figure 5A-5D.” (highlighted in yellow) in the figure legend of the new **Figure 5**. We converted the original Figure 5F to **Supplementary Figure S11**.

Original Figure 5

New Figure 5

Supplementary Figure S11

5.2. is there any information as to the litter size of the offspring or the total number of females that got pregnant in the PBS versus the Z2 receiving mice?

Response: Yes. The average litter size of the offspring of PBS-treated pregnant mice was 10.4 ± 1.5 , while that of the pregnant mice treated with Z2 at dose of 10, 20, 40, 80, and 120 mg/kg was 10.2 ± 1.6 , 10.5 ± 2.0 , 10.4 ± 2.1 , 10.6 ± 1.1 , and 10.8 ± 2.6 , respectively. We have added this information in the legend of **Figure 5 (lines 683 - 685)** :

Figure 5. Safety analysis of Z2 for pregnant ICR mice and fetuses. (A) Body weight changes of moms at different prenatal and postnatal time points. Thirty-one pregnant ICR mice (E12-14) were assigned randomly to six groups and were injected intravenously with PBS (n=5), or PBS containing Z2 at escalating dose (10 mg/kg, n= 5; 20mg/kg, n= 6; 40mg/kg, n= 5; 80mg/kg, n= 5; 120 mg/kg, n= 5) every day for 3 consecutive days. **(B)** Body weight changes of pups at various postnatal time points. The average litter size of the offspring of PBS-treated pregnant mice was 10.4 ± 1.5 , while that of the pregnant mice treated with Z2 at dose of 10, 20, 40, 80, and 120 mg/kg was 10.2 ± 1.6 , 10.5 ± 2.0 , 10.4 ± 2.1 , 10.6 ± 1.1 , and 10.8 ± 2.6 , respectively.

Reviewer #3 (Remarks to the Author):

Summary:

The authors present a new peptide drug (Z2 with 33 amino acids) with efficacy against Zika virus (ZIKV), which places a particular threat on pregnant women and, more specifically, on the developing fetus via teratogenic effects. The synthetic peptide Z2 is derived from the stem region of ZIKV envelope protein and is able to penetrate the placental barrier of pregnant, non-infected mice. Conversely, daily injection of Z2 protected non-pregnant mice from a lethal infection challenge with ZIKV. As other peptide drugs were previously used in pregnant women to prevent HIV transmission to the child and based on the presented data, this concept seems promising. As mode of action, the manuscript shows that Z2 disrupts the membrane of the enveloped ZIKV, leads to the release of viral genomes, and thus abrogates infectivity. The used methods are well suited to address the raised research questions, and the experiments are well controlled, including unrelated peptides or viruses. Nevertheless, several key points on the route of virus challenge and safety profile need to be addressed as outlined below before the manuscript may be suited for publication in Nature Communications.

MAJOR COMMENTS

1. The results that Z2 reaches fetuses of non-infected A129 mice (IVIS imaging of fluorescently labeled Z2, Fig. 4) seem disconnected from the data, showing that Z2 can protect adult not pregnant mice from ZIKV infection (survival data, Fig. 6). A major claim of the manuscript is that Z2 can be used during pregnancy to protect fetuses from ZIKV infection, but this claim is actually not addressed by the data. The authors need to

provide evidence that Z2 can prevent teratogenic effects in pregnant A129 mice by protecting the developing fetuses from ZIKV infection. These experiments could be performed using a lower and sublethal dose of ZIKV in pregnant mice and determine fetal development.

Response: We thank the reviewer for the constructive comments. We previously did plan to investigate whether Z2 could prevent teratogenic effects in pregnant A129 mice by protecting the developing fetuses from ZIKV infection. However, because of the difficulty in obtaining enough pregnant A129 mice (at least 10, E12-14), as noted above, we could not perform the necessary experiments before submission of our original manuscript. Fortunately, Dr. Cheng-Feng Qin (co-corresponding author) and his colleagues have recently established a C57BL/6 pregnant mouse model to study ZIKV vertical transmission ⁵. Therefore, we were able to use this mouse model to determine whether Z2 is able to block transmission of ZIKV transplacentally. As shown in new **Figure 6**, treatment with Z2 could significantly decrease the viremia of pregnant C57BL/6 mice and reduce the rate of ZIKV infection of placentas and fetal heads. Please see our response to Reviewer #2's critique (#1.2) for more details.

2. Fig. 5: The authors claim that Z2 is a safe peptide drug candidate for use in pregnant women. Safety of Z2 was, however, only tested in pregnant mice the dose that protected non-pregnant mice from ZIKV infection (10 mg/kg) and a 2-fold higher dose (20 mg/kg). Even though not showing in vitro cytotoxicity for cell lines (Fig. S4) or mouse red blood cells (Fig. S4), the performed test in mice is not sufficient to conclude about toxicity in vivo. Along the same lines, the shift in fluorescence of E-protein negative cells in Fig. 3B may indicate a non-specific binding of Z2 to non-infected cells, and Z2 reaches non-infected fetuses (Fig. 4), suggesting binding to non-infected cells and potential off-target effects. Consequently, a dose-escalation study in pregnant mice needs to be performed to determine potential toxicity.

Response: We thank the reviewer for the suggestion. We have added experiments to determine the potential *in vivo* toxicity of Z2 at dose of 40, 80, and 120 mg/kg body of weight, to pregnant mice. As shown in new **Figure 5**, moms or pups in each group grew normally. Z2 treated groups comparing with PBS group showed no significant differences in ALT and creatinine in the blood, and histological sections of organs were normal without exception. These results suggest that Z2 at the dose as high as 120 mg/kg is safe for pregnant mice, which is 11-fold higher than that providing protection against ZIKV infection *in vivo*. Indeed, some non-specific binding of Z2 was observed. However, we believe that this very limited non-specific binding will not significantly affect the *in vivo* efficacy of Z2.

We have added the following statement in the revised manuscript (**lines 219-221**): “Overall, Z2 is safe for pregnant ICR mice and fetuses, even at the dose as high as 120 mg/kg of body weight, which is 11-fold higher than that providing protection against ZIKV infection *in vivo*”.

3. To test the efficacy of Z2 *in vivo*, mice were injected intraperitoneally with Z2 one hour after infection with ZIKV with the same route. In addition, *in vitro* experiments were performed via pre-incubating ZIKV and Z2 for 1.5 hours. These procedures do not reflect the natural events that occur during ZIKV transmission via infected mosquitoes. The manuscript would largely benefit from showing that virus inoculation and treatment is independent. The authors could for instance inoculate ZIKV intravenous, subcutaneous, or intradermal to model natural routes of virus transmission and then inject Z2 intraperitoneal.

Response: We agree that the virus and peptide should be injected through different routes. Most recently, Aliota et al. and Julander et al. have reported that AG129 mice (I/II interferon receptor-deficient) could be infected by ZIKV through footpad injection^{6, 7}. Since we cannot get AG129 mice from animal suppliers in China, we obtained AG6 mice (I/II interferon receptor-deficient, C57BL/6 background). It has been shown that AG6 mice can be used as the lethal mouse model for studying infection of Dengue virus.⁸ According to the reviewer’s suggestion, we inoculated ZIKV subcutaneously (a natural route of ZIKV transmission) and injected Z2 peptide intraperitoneally, to analyze the efficacy of Z2 in AG6 mice. As shown in **Figure 7C and 7D**, Z2 intraperitoneally administered was highly effective against the lethal challenge with ZIKV via a subcutaneous route in the mouse footpad.

Figure 7. Protective activity of Z2 against ZIKV infection in lethal mouse models. (A) Survival of ZIKV-infected A129 mice. A129 mice (4 weeks old) were infected with 1×10^5 PFU of ZIKV through the intraperitoneal injection (i.p.) route. After 1 h, mice were treated with Z2 (n=8) at 10 mg/kg of body weight, and vehicle (n=8) as control. Mouse survival was observed and recorded daily until 21 days post-infection. (B) Viral RNA load in sera of ZIKV-infected A129 mice. At day 2 post-infection, mice were retro-orbitally bled to measure viral RNA load in sera by RT-qPCR. (C) Survival of ZIKV-infected AG6 mice. AG6 mice (6 weeks old) were infected with 1×10^3 PFU of ZIKV via a subcutaneous route in the footpad. After 1 h, mice were treated with Z2 (n=6) at 10 mg/kg of body weight, and vehicle (n=6) as control. Mouse survival was observed and recorded daily until 21 days post-infection. (D) Viral RNA load in sera of ZIKV-infected AG6 mice. At day 2 post-infection, mice were retro-orbitally bled to measure viral RNA load in sera by RT-qPCR. Whiskers: 5-95 percentile. **, $p < 0.01$; ***, $p < 0.001$.

We have added the results in the revised manuscript (**lines 247-250**): “Similarly, treatment with Z2 protected 67% of AG6 mice from death caused by subcutaneous administration of ZIKV and significantly prolonged MST from 10 days to the end of the experiment ($p < 0.01$) (**Figure 7C**). The viral load in Z2-treated AG6 mice at 2 days post-infection was about 13-fold lower than that of vehicle-treated mice ($p < 0.01$) (**Figure 7D**).”

We have added the method in the revised manuscript (**lines 471-478**) : “Twelve 6-week-old AG6 mice were assigned randomly to two groups and infected with 1×10^3 PFU of ZIKV (SZ01) via a subcutaneous route in the footpad. Then, the infected mice were i.p. administered with Z2 at 10 mg/kg of body weight (n=6) or vehicle control (n=6) every day for 6 consecutive days post-infection. The observation and examination of AG6 mice were then performed as described above.”

4. Along the same line as the point above, testing the spread of Z2 to the fetus was performed via intravenous injection (line 677), but testing the efficacy of the drug against ZIKV infection was performed via i.p. injection. For testing the spread of Z2 to the fetus, the same route of inoculation should be performed as for testing the efficacy of the drug.

Response: Thanks for the suggestion. Accordingly, we have assessed the distribution of Z2 peptide after intraperitoneal injection. As shown in **Supplementary Figure 9**, the Z2-Cy5 fluorescence was seen in the uterus and fetus, suggesting that Z2 could penetrate the placental barrier and enter into the fetus. We have added the results in the revised manuscript.

Supplementary Figure 9: Ability of Z2 to penetrate the placental barrier of pregnant ICR mice. (A) Imaging of pregnant ICR mice treated with Z2-Cy5 or PBS by the IVIS® Lumina K Series III from PerkinElmer. Mice were injected intraperitoneally with 100 μg Z2-Cy5 or PBS (n=3) as control (for background fluorescence measurement), followed by imaging analysis. (B) Imaging of the uteruses from the pregnant mice. (C) Imaging of the fetuses (n=9) removed from uteruses. (D) The statistical analysis of results from (B) and (C). Data are means \pm SD. *, $p < 0.05$; ***, $p < 0.001$.

• Line 46-47: The authors should mention the Americas in addition to Africa and Asia-Pacific region as areas at risk for ZIKV transmission.

Response: Thanks, we have added the following statement “In addition to Africa and Asia-Pacific region, the Americas, particularly the South America, are areas at risk for ZIKV transmission.” (lines 44-45).

References for response letter

- Jiang, S., Lu, H., Liu, S., Zhao, Q., He, Y. & Debnath, A. K. N-substituted pyrrole derivatives as novel human immunodeficiency virus type 1 entry inhibitors that interfere with the gp41 six-helix bundle formation and block virus fusion. *Antimicrob Agents Chemother* **48**, 4349-4359 (2004).
- Watanabe, W., Konno, K., Ijichi, K., Inoue, H., Yokota, T. & Shigeta, S. MTT colorimetric assay system for the screening of anti-orthomyxo- and anti-paramyxoviral agents. *J Virol Methods* **48**, 257-265 (1994).
- Muller, J. A., *et al.* Development of a high-throughput colorimetric Zika virus infection assay. *Med Microbiol Immunol*, 10.1007/s00430-017-0493-2 (2017).
- Guo, J., *et al.* Aptamer-functionalized PEG-PLGA nanoparticles for enhanced anti-glioma drug delivery. *Biomaterials* **32**, 8010-8020 (2011).
- Wu, K. Y., *et al.* Vertical transmission of Zika virus targeting the radial glial cells affects cortex development of offspring mice. *Cell Res* **26**, 645-654 (2016).

6. Julander, J. G., *et al.* Efficacy of the broad-spectrum antiviral compound BCX4430 against Zika virus in cell culture and in a mouse model. *Antiviral Res* **137**, 14-22 (2017).
7. Aliota, M. T., Caine, E. A., Walker, E. C., Larkin, K. E., Camacho, E. & Osorio, J. E. Characterization of Lethal Zika Virus Infection in AG129 Mice. *PLoS Negl Trop Dis* **10**, e0004682 (2016).
8. Liu, J., *et al.* Flavivirus NS1 protein in infected host sera enhances viral acquisition by mosquitoes. *Nat Microbiol* **1**, 16087 (2016).

REVIEWERS' COMMENTS:

Reviewer #1 (Remarks to the Author):

No further comments

Reviewer #2 (Remarks to the Author):

The authors have addressed all of my concerns.

Reviewer #3 (Remarks to the Author):

The authors have adequately addressed all of the reviewers' concerns, have included a substantial amount of additional data, and adapted the Results and Discussion accordingly. These adaptations have largely improved the quality and impact of the presented manuscript.